# Technical note: Precipitation phase partitioning at landscape-to-regional scales

Elissa Lynn[1], Aaron Cuthbertson[1], Minxue He[1], Jordi P. Vasquez[1], Michael L. Anderson[1], Peter Coombe[1], John T. Abatzoglou[2], Benjamin J. Hatchett[3]

[1]California Department of Water Resources, Sacramento, California, 95814, USA
[2]Management of Complex Systems Department, University of California, Merced, Merced, California, 95340, USA
[3]Western Regional Climate Center, Desert Research Institute, Reno, Nevada, 89512, USA

*Correspondence to*: Benjamin J. Hatchett (Benjamin.Hatchett@gmail.com)

**Abstract.** Water management throughout the western United States largely relies on the partitioning of cool season mountain precipitation into rain and snow, particularly snow as it maximizes available water for warm season use. Recent studies indicate a shift towards increased precipitation falling as rain, which is consistent with a warming climate. An approach is presented to estimate precipitation phase partitioning across landscapes from 1948-present by combining fine scale gridded precipitation data with coarse scale freezing-level and precipitation data from an atmospheric reanalysis. A marriage of these datasets allows for a new approach to estimate spatial patterns and trends in precipitation partitioning over elevational and latitudinal gradients in major water supply basins. This product is used in California as a diagnostic indicator of changing precipitation phase across mountain watersheds. Results show the largest increases in precipitation falling as rain during the past seven decades in lower elevation watersheds located within the climatological rain-snow transition regions of northern California during spring. Further development of the indicator can inform adaptive water management strategy development and implementation in the face of a changing climate.

## 1 Introduction

Mountains are natural reservoirs of water for human and natural consumptive uses in many parts of the world (Huss et al., 2017). In snow-dominated mountain environments, substantial quantities of water stored as snow accumulates during the cool season and is released during the warm season as snow melts. The partitioning of precipitation into liquid (rain) and, in particular, frozen (snow) components along with climatic stationarity were foundational assumptions in the development of water management infrastructure and practices in California and other mountain environments in the western United States (US) since the mid-1800s (Milly et al., 2008). Precipitation phase partitioning during the cool season influences the timing and magnitude of surface runoff, evapotranspiration, and groundwater recharge (Berghuijs et al., 2014; Zhang et al., 2015; Musselman et al., 2017; Sturm et al., 2017; Abatzoglou and Ficklin, 2017). The phase of cool season precipitation ultimately drives water management strategies, especially in arid and semi-arid environments characterized by substantial interannual hydroclimate variability (e.g., Sterle et al., 2019).

Many historically snow-dominated mountains in the western US, particularly those with mild winter temperatures, are undergoing declines in snow accumulation (Mote et al., 2018). Projections for the 21[st] century suggest continued warming and snowpack declines (Rhoades et al., 2018a). Because of downstream dependence on snow-derived water resources and susceptibility to flooding from snow melt events, California is an ideal location to examine changes in historical precipitation partitioning. Studies have found evidence for changes in California's cryosphere consistent with a warming climate including an upslope shift in winter snow line elevation (Hatchett et al., 2017), delayed early season snowpack accumulation (Hatchett and Eisen, 2019), earlier peak

snow water equivalent (Kapnick and Hall, 2010), and decreased snowpack water storage efficiency as measured by ratios of spring snow water equivalent to cool-season precipitation accumulation (Das et al., 2009).

Decreases in snowpack and snow-covered area exacerbates snow loss through the snow-albedo feedback (Walton et al., 2017). This effect is pronounced in lower elevation, warmer regions of watersheds where snow cover tends to be shallower and more ephemeral. The effects of a warming climate on snowpack vary, with the greatest sensitivity found in warm snow climates located near the climatological rain-snow transition elevation (Howat and Tulaczyk, 2005; Mote et al., 2005), predisposing these regions to warming-induced hydrologic vulnerability (Huning and AghaKouchak, 2018; Klos et al., 2014; Rhoades et al., 2018a). Changes in rain-snow partitioning and its manifestation on water storage in spring snowpack are thus of paramount importance to guiding changes in water resource management operations intended to enhance water supply reliability.

The sparse observational networks and complex topography of the western US introduces challenges into basin-scale hydrologic monitoring and modelling. To address these challenges when applying hydrologic models or for monitoring long-term change, the incorporation of multiple sources of data (Bales et al., 2006) and/or model output (e.g., Wrzesian et al., 2019) is often required. Dataset inadequacies have limited the use of precipitation partitioning for operational purposes as readily-available metrics are provided at scales too coarse for decision making processes or involve observational records that are limited temporally (e.g., < 30-year records) for climatological context. Daily gridded products based on sparse observational networks in mountainous areas have their own suite of limitations, such as resolving lapse rates that lead to challenges in near-surface temperature estimates (Lute and Abatzoglou, 2020). These are among many inadequacies regarding datasets or climate metrics faced by water managers (e.g., Jagannathan et al., 2020). To overcome such limitations, the California Department of Water Resources (DWR) developed a methodology to study historical precipitation partitioning trends at spatial scales relevant to broader management goals and capable of resolving finer scale details across elevational and climatic gradients (DWR, 2014). This technical note describes the development of this diagnostic indicator aimed at quantifying how rain and snow are partitioned at actionable scales for water management by integrating meteorological datasets.

Since 2015, DWR has documented this indicator in its annual Hydroclimate Report (DWR, 2019). Though not used directly in operational forecasts, the indicator provides DWR with situational awareness regarding the location of changes in precipitation phase and the rates of these changes. While we focus on California watersheds, an example application to the western United States is also provided. We suggest that this approach is scalable to regional-to-continental scales and therefore could be an informative diagnostic for water resources management and model development in other snowmelt dependent regions. Last, the methodology can also be applied to climate model projections to help inform the development of adaptation strategies to achieve water resource management goals amidst a changing climate.

## 2 Data

### 2.1 Study Areas

The study areas encompass the Sierra Nevada and Southern Cascades of California (Fig. 1a), with the middle and upper elevations historically receiving the majority of cool season precipitation as snow (Fig. 1c). Runoff originating from melting snow in these regions provides critical water resources for local, state, and federal water projects in California (Kahrl, 1979). Guided by hydroclimate conditions, such as accumulated winter precipitation and the spatial distribution of snow water equivalent (SWE; the amount of liquid water stored in the snowpack), DWR produces monthly forecasts of unimpaired April through July runoff

forecasts beginning in early February and published as Bulletin 120 (DWR, 2019). The forecasts in Bulletin 120 are updated weekly until June as conditions evolve. Based upon DWR's management of state water resources, these snowpack-dominated mountain areas are subset into four analysis zones (from north to south: Southern Cascades, Northern, Central, and Southern Sierra Nevada; Figure 1a). The elevation distribution of the analysis zones shifts higher with decreasing latitude. Median annual precipitation is the greatest in the higher latitude Northern Sierra Nevada and Southern Cascade regions. In total, 33 United States Geological Survey eight-digit Hydrologic Unit Code (HUC-8) watersheds are included within the four analysis zones. The study period spans water years (WY) 1949-2018. A water year begins on 1 October of the prior calendar year and ends on 30 September.

## 2.2 Data used in DWR approach to rain/snow partitioning

The DWR approach uses monthly, 800 m horizontal resolution estimates of precipitation from the Parameter Regression Interpolated Slopes Model (PRISM; Daly et al., 2008), a digital elevation model (DEM) corresponding to the PRISM grid, and freezing level elevations from the North American Freezing Level Tracker (described in section 2.3). The method produces watershed-aggregated monthly time series of total precipitation and percentage total precipitation estimated as snow ($\%_{SNOW}$). These time series are analysed for the entire water year (October-September), fall (September-November), winter (December-February), and spring (March-May). Because the 800 m PRISM products are not freely available to the public, we use the 4 km monthly products spanning 1948-present from the PRISM group (http://www.prism.oregonstate.edu/).

## 2.3 The North American Freezing Level Tracker

The North American Freezing Level Tracker (NAFLT, https://wrcc.dri.edu/cwd/products/) was developed by the Western Regional Climate Center in 2008 to provide estimates of the height of the freezing level, or elevation of the 0°C isotherm, across North America, based upon 6-hourly output from the National Center for Environmental Prediction/National Center for Atmospheric Research Global Reanalysis spanning 1948-present at a 2.5° horizontal resolution (hereafter "NCEP/NCAR reanalysis"; Kalnay et al., 1996). The height of the freezing level is an important parameter for evaluating climate variability and change in mountain environments (Diaz et al., 2003). Freezing level height influences the phase of precipitation at a given elevation, the state of the land surface (frozen or un-frozen), the thermodynamic processes occurring in an existing snowpack leading to snowpack ripening and melt, and the duration of the snow-free season (Diaz et al., 2003; White et al., 2010; Sospedra-Alfonso et al., 2015; Contosta et al., 2019).

The NAFLT calculates the freezing level as the highest elevation in the troposphere (200-1000 hPa) above mean sea level where free-air temperatures are 0°C for each 2.5° NCEP/NCAR grid point (Step 1 in the conceptual diagram shown in Figure 2). If the entire atmosphere is at or below freezing on a given 6-hr period, a value of zero meters above mean sea level is provided. For cases in which the vertical temperature profile includes inversion conditions with multiple incursions of the 0°C isotherm, the uppermost atmospheric level below which the 0°C isotherm occurs is used. In addition to providing estimates of the elevation of the 0°C isotherm, the NAFLT calculates the monthly percent of precipitation that falls as snow ($\%_{SNOW}$) at 200 m elevational increments from 0-4000 m. This is done by assigning all 6-hourly modelled precipitation from the NCEP/NCAR reanalysis as snow for elevations above the corresponding freezing level and all precipitation in a 6-hour increment as rain for elevations below the freezing level (Steps 2 and 3 in the conceptual diagram). The freezing level is a conservative estimate of the snow level as precipitation can often persist as snow below the freezing elevation due to latent heat fluxes (e.g., snow falling in a sub-saturated atmosphere, deep isothermal temperature profiles, or during heavy precipitation that entrains colder air and drags it downward to

lower levels in the atmosphere; Minder et al., 2011; Jennings et al., 2018). However, accumulations of snow below the elevation of the 0°C isotherm may be transient due to nominal cold content of snow.

## 3 Methods

### 3.1 Description of the DWR approach to rain/snow partitioning

The DWR approach calculates %$_{SNOW}$ by first bilinearly interpolating the 2.5° grid point estimates of %$_{SNOW}$ horizontally for each 200 m elevational increment from the NAFLT (Step 4 in the conceptual diagram). The approach next assigns %$_{SNOW}$ to each fine-scale (4 km) PRISM grid point per the smallest elevational difference between fine-scale elevation (e.g., 4 km DEM) and the 200 m elevational bins. If the freezing level elevation is below the terrain elevation, all precipitation falls as snow (%$_{SNOW}$ = 100%). Given the inadequacies of coarse-scale reanalysis precipitation fields, when calculating seasonal totals, we multiplied estimates of
monthly PRISM precipitation by monthly %$_{SNOW}$ to partition precipitation between total frozen (%$_{SNOW}$) and liquid (%$_{RAIN}$) components. We then sum over the months to calculate the seasonal or water year %$_{SNOW}$ using the PRISM-weighted precipitation estimates. We report %$_{SNOW}$ using the seasonal or water year ratio of frozen water to liquid water.

The statewide, analysis zone, and watershed average annual precipitation and total average annual %$_{RAIN}$ (or %$_{SNOW}$) can be
calculated by aggregating data at the native resolution (e.g., 4 km) to the spatial unit of analysis, such as a watershed. These metrics are reported annually by DWR in annual hydroclimate reports. As examples, Figure 1 allows comparisons between three different water years, a record low snowpack year (2015; Fig. 1d), a near-average year (2008; Figure 1e), and a year with much higher partitioning of precipitation as snow (1980; Fig. 1f). State-wide 1 April SWE in 2015 was the lowest since DWR began record-keeping in 1929, while both 2008 and 1980 had SWE values near the long-term 1 April average.


The methodological approach of the NAFLT assumes that freezing levels at the chosen analysis points are representative of synoptic scale weather conditions. Despite known mesoscale variability in snow line elevation during individual events (e.g., Minder et al., 2011), reasonably little bias in snow levels (at the interannual timescale) exist between stations located within 200 km of one another along the windward side of the Sierra Nevada (Hatchett et al., 2017). Thus, the 2.5° (~280 km) horizontal
resolution of NAFLT appears reasonable for the purpose of interannual tracking of rain/snow partitioning. By performing calculations of precipitation phase at 6-hourly intervals, our method is better able to capture changes in freezing level and its impact on precipitation (e.g., frontal passage) than daily approaches that can smooth out these influences.

### 3.2 Statistical Analysis

Temporal trends in historical rain/snow partitioning were evaluated spanning water years 1949-2018 using the non-parametric
Mann-Kendall test modified to account for temporal autocorrelation (Hamed and Rao, 1998). Significance was determined using an alpha level of 0.05. When noted, only grid points with statistically significant trends are shown in the resulting figures, with all trends provided in the supplementary information. Trends were calculated by multiplying the Theil-Sen slope by 10 (yielding change in %$_{SNOW}$ decade$^{-1}$) at each 4 km grid point for late fall (October-November), meteorological winter (December-February), early spring (March-April), and the cool season (October-April). These calculations were performed over the western United
States, though we constrain most of our focus to the Sierra Nevada and Southern Cascades of California. To highlight the spatial information provided by the approach, we also calculated trends aggregated by latitude and elevation across the area within the four analysis zones over the cool season and also at HUC-8 watershed scales. For the watershed-level aggregations, a precipitation-weighted average %$_{SNOW}$ was calculated over the area within a given watershed and the trend calculation was then performed. To

account for precipitation heterogeneity within watersheds, we calculated watershed %$_{SNOW}$ by separately summing the total frozen precipitation and total precipitation across all grids within a watershed and dividing the two.

## 4 Results

Trends in estimated changes in %$_{SNOW}$ (shown as % decade$^{-1}$) for winter (Fig. 3a), spring (Fig. 3b), and the cool season of the water year (Fig. 3c) range from no change in the highest elevations of the central and southern Sierra Nevada (and Mt. Shasta in the southern Cascades) to decreases of 4% decade$^{-1}$ in lower and middle elevation regions over the 70-year record. Winter season trends were largest in the southern portion of the northern Sierra Nevada region and throughout the central Sierra Nevada region and on the order of -1% to -2% decade$^{-1}$. Spring trends were of larger magnitude (-2% to -4% decade$^{-1}$) and concentrated in the middle elevations of all regions. The highest elevations of the southern Sierra Nevada showed no declines as these locations remain upslope of the 0°C elevation during these seasons. Fall trends (not shown) were negative, but magnitudes were smaller than winter trends. No statistically significant positive trends were observed for any season.

Trends at the HUC-8 watershed scale show similar results (Fig. 4). The largest negative changes are found in the central Sierra Nevada region on both westward and eastward draining watersheds (i.e., west and east of the Sierra Nevada crest, respectively). These areas show the greatest magnitudes of change at middle elevations during the spring (Fig. 4b). Fall and winter trends moderate the magnitudes of the cool season trends (Figure 4c).

Trends in %$_{SNOW}$ exhibit strong spatial patterns than can further be explored and understood by binning trends by elevation. The largest negative trends in water year partition of precipitation as snow across the four regions were seen at mid-elevations of 1800-2500 m (-1.5 to -2% decade$^{-1}$) and become notably weaker at higher elevations that are climatologically well above the 0°C elevation during winter months (Fig. 5a). Lower elevations (<1800m) occupy a larger portion of the collective watershed area and had significant declines in %$_{SNOW}$ (-1 to -1.5% decade$^{-1}$). Further decomposition of trends by elevation and latitude shows the largest declines in %$_{SNOW}$ at mid-elevations in the southern extent of the region (Fig. 5b), consistent with Figure 4. However, we note the strongest negative trends south of 38°N occupy a much smaller geographic extent of overall watersheds than those located further north in California.

Long-term trends throughout the western United States (Fig. 6) demonstrate similar magnitudes of change as found in California with decreases on the order of -0.5% to -4% decade$^{-1}$. A trend towards less precipitation as snow during fall in the higher elevations is noted in the Rocky Mountains in Colorado and northwestern Montana as well as the Wind River Range in Wyoming (Fig. 6a; Supplementary Figure 2). Areas east of the Cascade Range (central and northern Washington and central Oregon), the Montana plains, western and northern New Mexico, and much of the non-mountainous terrain in Wyoming and in the Colorado River Basin show the greatest magnitudes of decreases in winter %$_{SNOW}$ (Fig. 6b). As we found in California, the spring season showed the largest magnitudes of decreases in %$_{SNOW}$ (Fig. 6c) with the greatest magnitudes in central Nevada, southwestern Utah, central Arizona, and along the Front Range of the Colorado Rockies. Averaged over the cool season, the western United States demonstrated decreases in %$_{SNOW}$ by approximately -1% to -2% decade$^{-1}$ over the past ~70 years (Fig. 6d).

# 5 Discussion

## 5.1 Is there a transition to "more rain, less snow?"

Combining 4 km PRISM monthly precipitation and using freezing level estimates from reanalysis confirms widespread declines in the percent of precipitation falling as snow over California (Fig. 4) and the western United States (Fig. 6). The most notable, or largest magnitude, and widespread changes have occurred in spring at elevations near and below the climatological 0°C height. The apparent asymmetric warming of the leeside of the Sierra Nevada compared to the windward side (Fig. 3) warrants additional investigation to elucidate physical mechanisms generating this asymmetry. The watershed-scale signal (Fig. 4) may also be a by-product of the greater land area at middle elevations in leeside watersheds where trends have the greatest magnitudes (Figure 5a). A benefit of the spatially distributed nature of the DWR approach is that it facilitates the identification of spatial behaviours that may not be readily apparent using sparsely distributed station observations.

The method presented agrees well with previous station-based observations showing declines in %$_{SNOW}$ (e.g., Knowles et al. 2006). The gridded nature of the approach used allows detailed analyses at the regional or watershed level, both spatially (Fig. 4) and across binned elevations and latitudes (Fig. 5) that adds nuance to the analysis. In the case presented, the aggregation techniques highlight the magnitude of change as a function of elevation and latitude (Fig. 5a) to elucidate hydrologic basins that may be most susceptible to changes in precipitation partitioning (Fig. 4).

The spring season signal of increasing precipitation as rain, especially in middle elevation zones and southern upper elevation zones of California and throughout much of the western United States, is consistent with declines in peak snowpack, changes in plant phenology, and earlier timing of runoff (Cayan et al., 2001; Das et al., 2009; Kapnick and Hall, 2010; Mote et al., 2018). Snowpack declines are robustly projected to continue into the 21st century (Rhoades et al., 2018b) and be further exacerbated during droughts (Berg and Hall, 2017) and extreme wet years (Huang et al., 2018). The method presented also suggests that the highest elevation regions in the Sierra Nevada, the Wasatch Range, and the Rocky Mountains have not experienced significant declines in precipitation falling as snow to date during winter and spring. With continued warming and increased freezing levels, however, these areas are posited to undergo declines in %$_{SNOW}$ (Klos et al., 2014; Huang et al., 2018; Rhoades et al., 2018b; Sun et al., 2019).

The transition from snow to rain at lower and middle elevations of California's Sierra Nevada during the primary accumulation seasons (Fig. 5a-b) has reduced the amount of water stored as spring snowpack (Mote et al., 2018). This declining capability of mountains to act as natural reservoirs is a key response to climate warming (Rhoades et al, 2018a). It has also led to more frequent warm snow drought conditions (Hatchett and McEvoy, 2018). More precipitation falling as rain during storms, especially in regions with large watershed areas in lower elevations, increases midwinter inflow into reservoirs. Many current multipurpose reservoir management paradigms require the maintenance of a flood pool, which is reservoir storage space allocated to attenuate periods of heavy inflow and reduce flood hazard during cool season storms. Water captured during the flood is later released to maintain the flood pool storage capabilities during the next possible event. Flood pool releases mean this water cannot be stored for later beneficial use and must be managed as a hazard rather than a resource. Work is in progress to develop adaptation strategies such as forecast-informed or dynamic reservoir operations (Steinschneider and Brown, 2012; Talbot et al., 2019) and managed aquifer recharge (e.g., Dillon et al., 2010) to address this growing water management challenge as continued warming results in additional changes from snow to rain. In watersheds with minimal or no reservoir storage, changes from snow to rain may have more impactful changes on flood hazard and habitat, especially during low warm season flows, thus requiring more creative or costly solutions.

Other non-traditional strategies to offset projected decreases in mountain snowpack and achieve water supply reliability exist, such as storm water recapture, water recycling, and water markets. However, these will require economic assessments to determine feasibility (Cooley et al., 2019).

**5.2 Primary Limitations**

The approach described herein does have several primary limitations in its current form. A major limitation is the assumption that the NAFLT freezing level elevation linearly corresponds to the %$_{SNOW}$ estimate, which is then multiplied by the PRISM precipitation amount at that grid point at the monthly time scale. One key limitation of PRISM in this application is that it remains an interpolation method based on *in situ* observational data, which is sparse in mountainous regions (Henn et al., 2018). Indeed, some high-resolution model simulations show more realistic precipitation amounts in mountains than some observational networks

(Lundquist et al., 2020; Wrzesien et al., 2019). At the watershed scale, differences between PRISM products (i.e., 4 km, 800 m) and their associated elevation for prescribing local %$_{SNOW}$ is likely nominal. However, we would expect site-specific comparisons to yield differences that may be of importance for smaller watersheds and ecological processes.

Our assumption that coarse models (e.g., reanalysis products) accurately represent the freezing level ignores mesoscale effects of

snowline variability in complex terrain (Minder et al., 2011) and the effects of near-surface humidity (Harpold et al., 2017). Both sources of uncertainty may result in substantial biases in rain/snow partitioning estimates as a function of individual storms, particularly during frontal passage and when the magnitude and spatial distribution of precipitation is also considered. Further, comparisons with approaches that include relative humidity or wet bulb temperatures are recommended to further improve the methodology, as these have been shown to improve the quality of rain-snow partitioning (Harpold et al., 2017, Wang et al., 2019).


The NCEP/NCAR reanalysis, which the NAFLT uses to identify freezing levels and partition precipitation, is an older generation reanalyses product. Recent advances in atmospheric reanalyses such as ERA-5 (Hersbach et al., 2020) provide advances in data assimilation procedures, have finer spatiotemporal resolution, and provide 0°C heights as standard products. A comparison of the NCEP/NCAR approach to ERA-5 during 1979-2018 showed strong similarity in the spatial distribution of %$_{SNOW}$ (Supplementary

Figure 3) and high interannual correlations ($0.9 < R < 0.99$), with slightly higher %$_{SNOW}$ in ERA-5 (Supplementary Figure 4). The method for partitioning precipitation described herein shows promise using the older NCEP/NCAR reanalysis, but it flexible enough to incorporate advances in reanalyses products as well as climate model projections.

**6 Concluding Remarks**

Changes in the fraction of precipitation falling as snow during the cool season can have significant impacts on the ability of water

managers to balance management objectives (e.g., water supply, ecosystem demands, and recreation) through reservoir operations. Expectations from climate change projections suggest that dynamic adaptation strategies will have to be employed to maintain the functionality of existing water management infrastructure. These strategies will rely on managers having estimates of spatially distributed historical precipitation phase partitioning at landscape scales readily available for use. We presented a method for estimating snowfall as a fraction of total precipitation at high spatial resolution (e.g., 4 km) and modest temporal resolution

(monthly) with output from the North American Freezing Level Tracker (NAFLT) based on a global reanalysis product (NCEP/NCAR), PRISM precipitation, and a digital elevation model. A trend analysis indicates a greater fraction of precipitation across California's historically snow-dominated mountain regions with the spring showing the strongest trends (-2% to -4% decade$^{-1}$) followed by winter (-1% to -2% decade$^{-1}$). The largest decreases were found at mid-elevations near the climatological freezing

level, which have previously been identified as the most vulnerable to warming (Huning and Aghakouchak, 2018). These products provide complementary information to high resolution snow reanalyses that incorporate satellite and/or *in situ* data (e.g., Margulis et al., 2016; Zeng et al. 2018).

The developed method uses publicly-available gridded data sets that enable application to areas with similar natural resource or water management paradigms. Ongoing work seeks to address the limitations presented in order to produce more robust estimates of historical change in rain/snow partitioning and enable additional storm or place-based detail that can be utilized in adaptive strategy development and applications. The main advantage of the described approach is that the NAFLT can be periodically updated as higher resolution gridded data products become available, including those at global scales (e.g., TerraClimate;

Abatzoglou et al., 2018) and global and regional climate models. Further examination of how freezing levels are influenced by large scale modes of climate variability are also recommended. For example, Abatzoglou (2011) found trends in the Pacific-North American pattern contributed to increases in freezing levels and declines in precipitation falling as snow. Evaluating freezing level and precipitation phase relationships to isolated modes of climate variability may provide useful guidance for hydroclimate predictability at lead times relevant for water management (e.g., Patricola et al. 2020).

It is anticipated that an updated freezing level tracker tool will be developed and used to provide precipitation phase partitioning information to water managers to help inform decision making. California's investment in unique data sets like snow level radar (White et al., 2013) coupled with ongoing efforts to improve *in-situ* weather monitoring in headwater regions (Lundquist et al., 2016) creates an opportunity for further exploration of rain/snow partitioning including storm-based and place-based analyses.

These analyses can play important roles in developing and implementing adaptive strategies for water management by providing analogues to future cool seasons or storm events in a warming climate (e.g., Berg and Hall, 2017; Hatchett, 2018; Huang et al., 2019; Sterle et al., 2019).

## 7 Acknowledgements

This work was supported by the California Department of Water Resources. We dedicate this method to the late Dr. Kelly T. Redmond, who always encouraged the application of climate science to inform decision making in the western United States. We appreciate the constructive reviews provided by Alan Rhoades and an anonymous reviewer.

## 8 Code/Data availability

The processing code, and processed data (e.g., %$_{SNOW}$) is available upon request.

## 9 Author contributions

Elissa Lynn, Aaron Cuthbertson, Minxue He, Jordi P. Vasquez, Michael L. Anderson, and Peter Coombe conceptualized the idea. Elissa Lynn supervised the project. Benjamin J. Hatchett wrote the manuscript with input from all authors and was responsible for

analysis and visualization. John T. Abatzoglou developed the North American Freezing Level Tracker, generated the precipitation phase partitions, and performed the analysis and visualization shown in Figure 4. All authors contributed to the interpretation and presentation of data and results as well as the revision and editing of the original manuscript.

## 10 Competing interests

Authors Elissa Lynn, Aaron Cuthbertson, Minxue He, Jordi P. Vasquez, Michael L. Anderson, and Peter Coombe are employed by the California Department of Water Resources. Authors Benjamin J. Hatchett and John T. Abatzoglou declare that they have no competing interests.

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

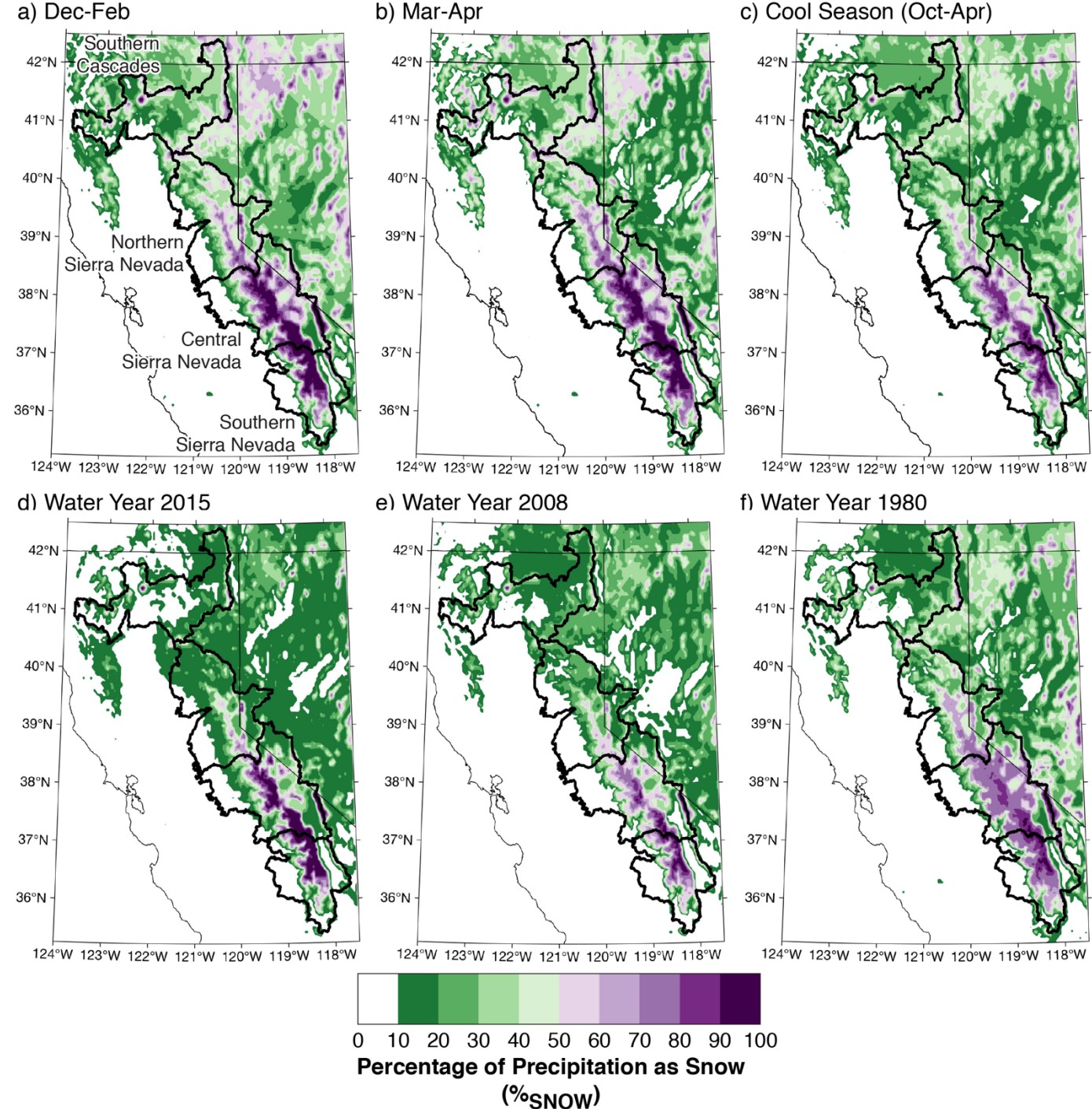

**Figure 1: Estimated historical (1950-1969) percentages of precipitation as snow for (a) winter (Dec-Feb), (b) spring (Mar-Apr), and (c) for the full cool season (Oct-Apr). Examples of %$_{SNOW}$ averaged over the cool season (October-April) of water years (d) 2015, (e) 2008, and (f) 1980. Thick black contours denote California Department of Water Resources analysis zones.**

Step 1 (N. American Freezing Level Tracker): Working downward from 200 hPa, identify height of freezing level for each NCEP/NCAR grid cell at each 6 hr time step via linear interpolation, assign to 200 m bin (from 0 - 4000 m)

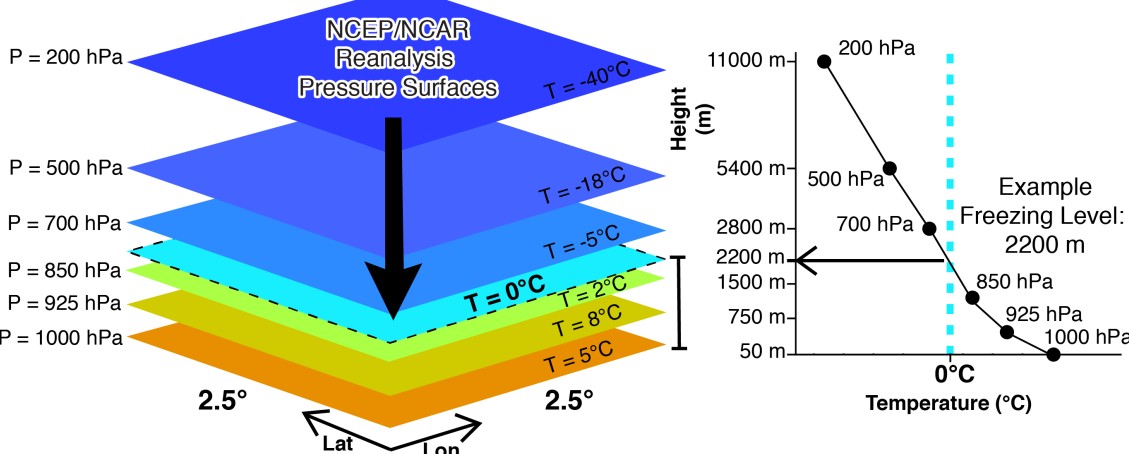

Step 2: Assign phase to precipitation (P) based on freezing level for bins above and below, where precipitation as snow = Psnow and precipitation as rain = Prain

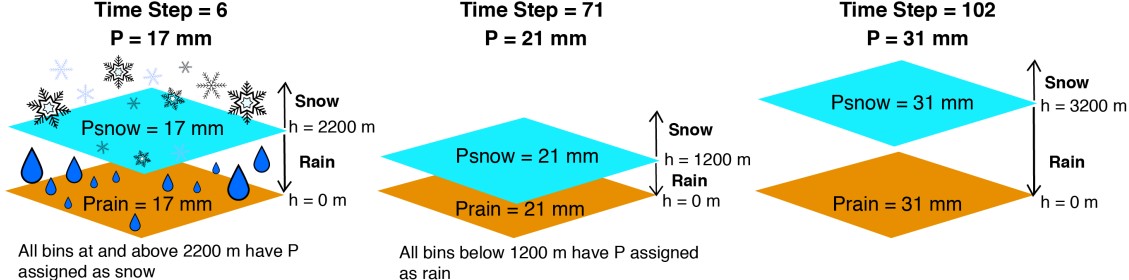

Step 3: For each 200 m elevation bin at each grid point, sum Psnow Prain over all time steps in the month then divide by total P to calculate monthly %SNOW

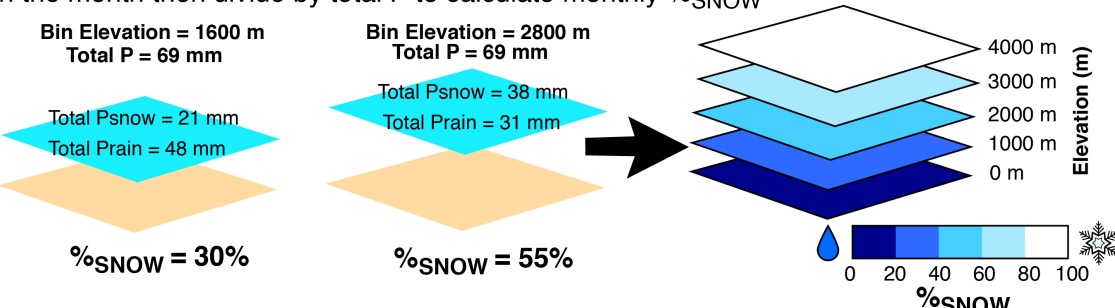

Step 4: (DWR Method) Distribute %SNOW over the 4 km PRISM grid points (binned by 200 m intervals) using bilinear interpolation. For seasonal values, use PRISM P to scale %SNOW

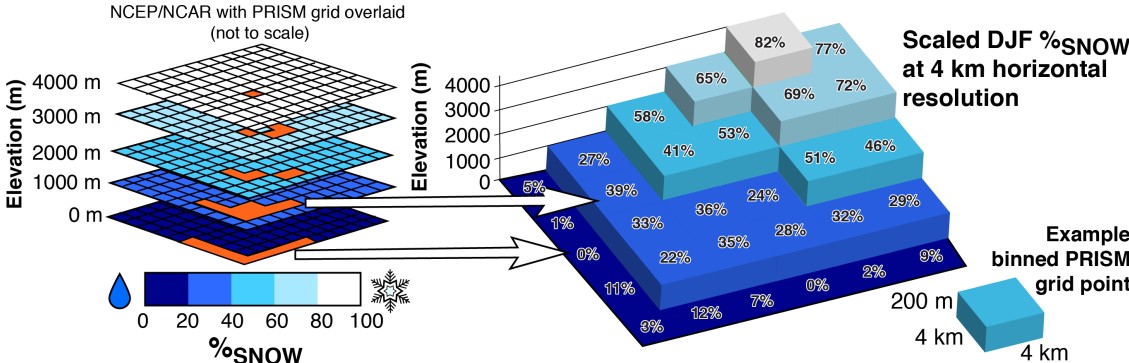

**Figure 2: Conceptual diagram illustrating the four key steps in the calculation of %SNOW at 4 km horizontal resolution and using 200 m elevation bins starting with 2.5° x 2.5° horizontal resolution NCEP/NCAR reanalysis.**

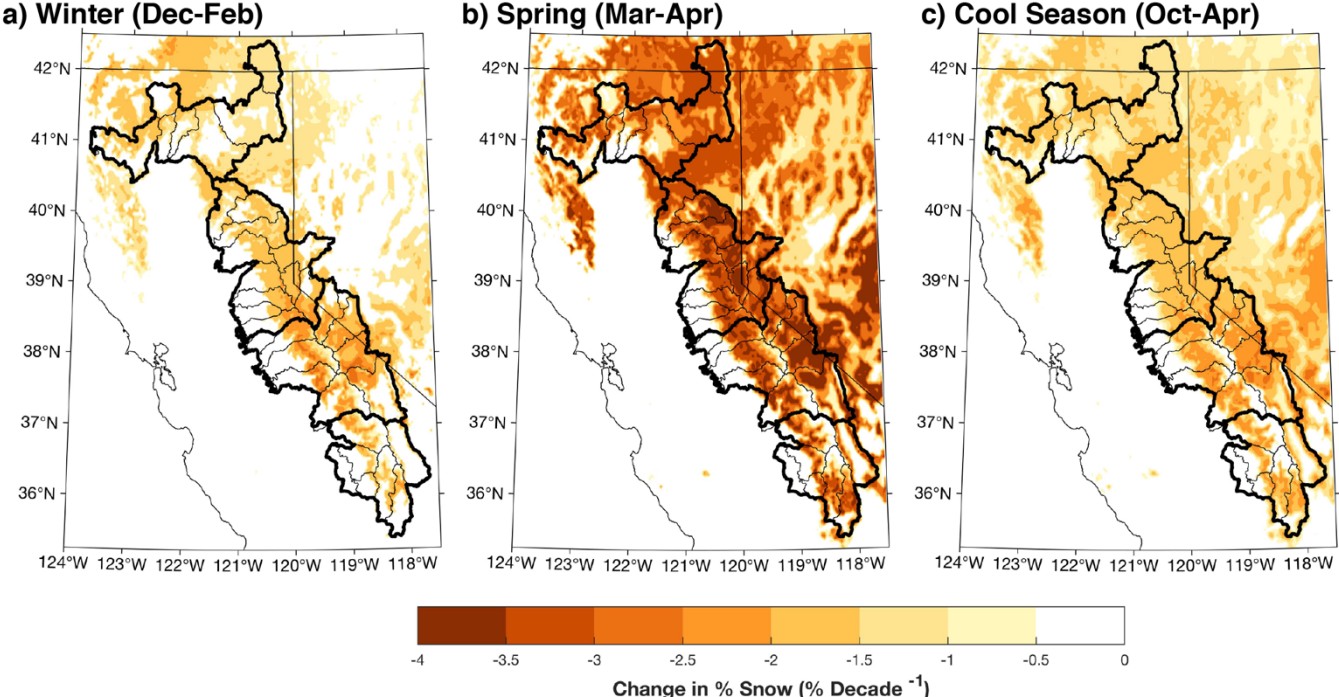

**Figure 3: Estimated changes in %$_{SNOW}$ (in % decade$^{-1}$) for (a) winter (Dec-Feb), (b) spring (Mar-Apr), and (c) for the full cool season (Oct-Apr). Thick black contours denote California Department of Water Resources analysis zones. Thin black contours denote United States Geological Survey HUC-8 watersheds. Only gridpoints with statistically significant (p<0.05) trends are shown; Supplementary Figure 1 shows trends for all gridpoints.**

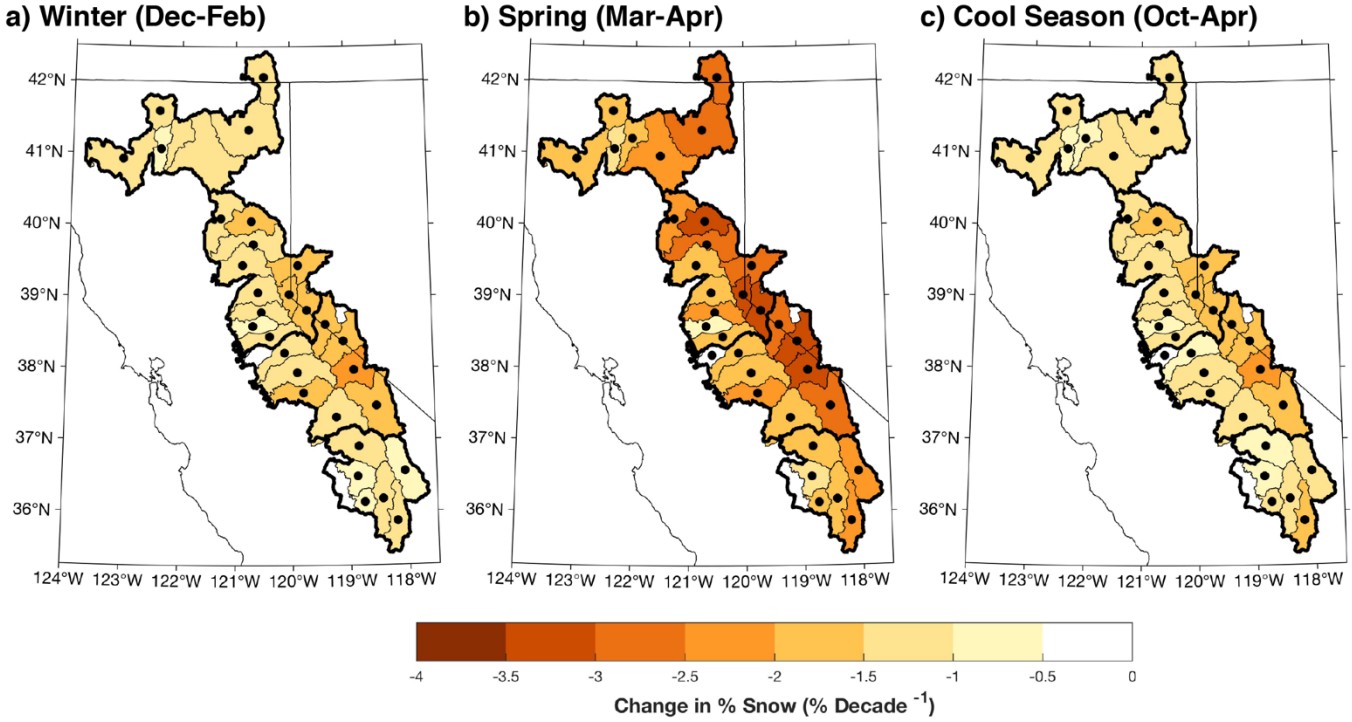

**Figure 4: As in Figure 3 but with trends averaged over HUC-8 watersheds. Filled black circles indicate statistically significant (p<0.05) trends.**

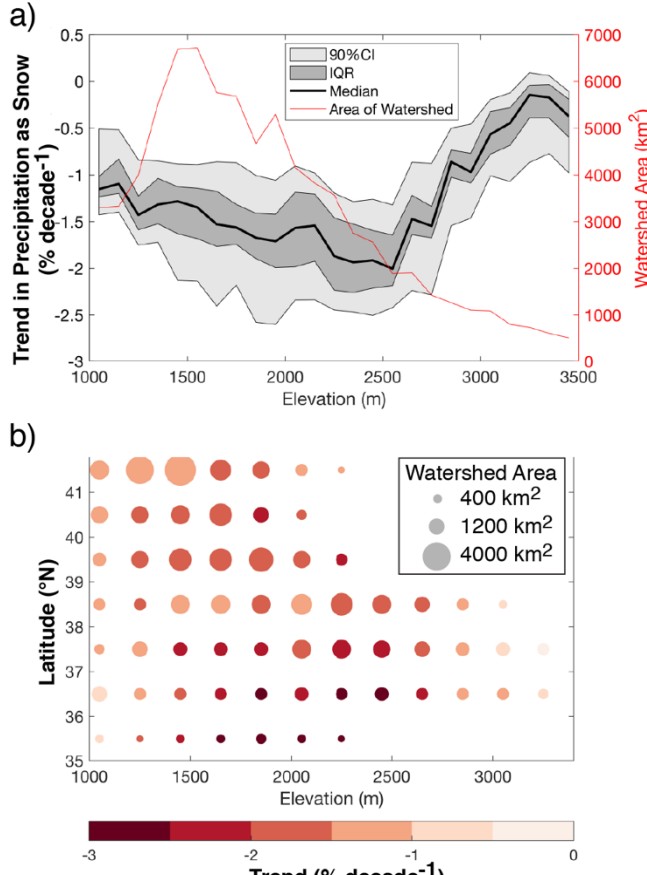

**Figure 5: (a)** Elevation-based trends (aggregated over all latitudes) of %$_{SNOW}$ (% decade$^{-1}$) showing median (black line), the interquartile range (dark grey shading), and 90% confidence intervals (light grey shading) on the left y-axis. Right y-axis shows the total watershed area occupied by each elevation bin (red line; km$^2$). **(b)** Aggregated trends in %$_{SNOW}$ (% decade$^{-1}$) by latitude and elevation for the water year. Dot size is scaled by area of watershed occupying each elevation and latitude bin. Aggregations were performed on gridpoints within the subset of California Department of Water Resources analysis zones (see Figure 1a) and sorted by elevation. The interquartile range (IQR) and 90% confidence interval (CI) were estimated using all grid points within each elevation band and analysis zone.

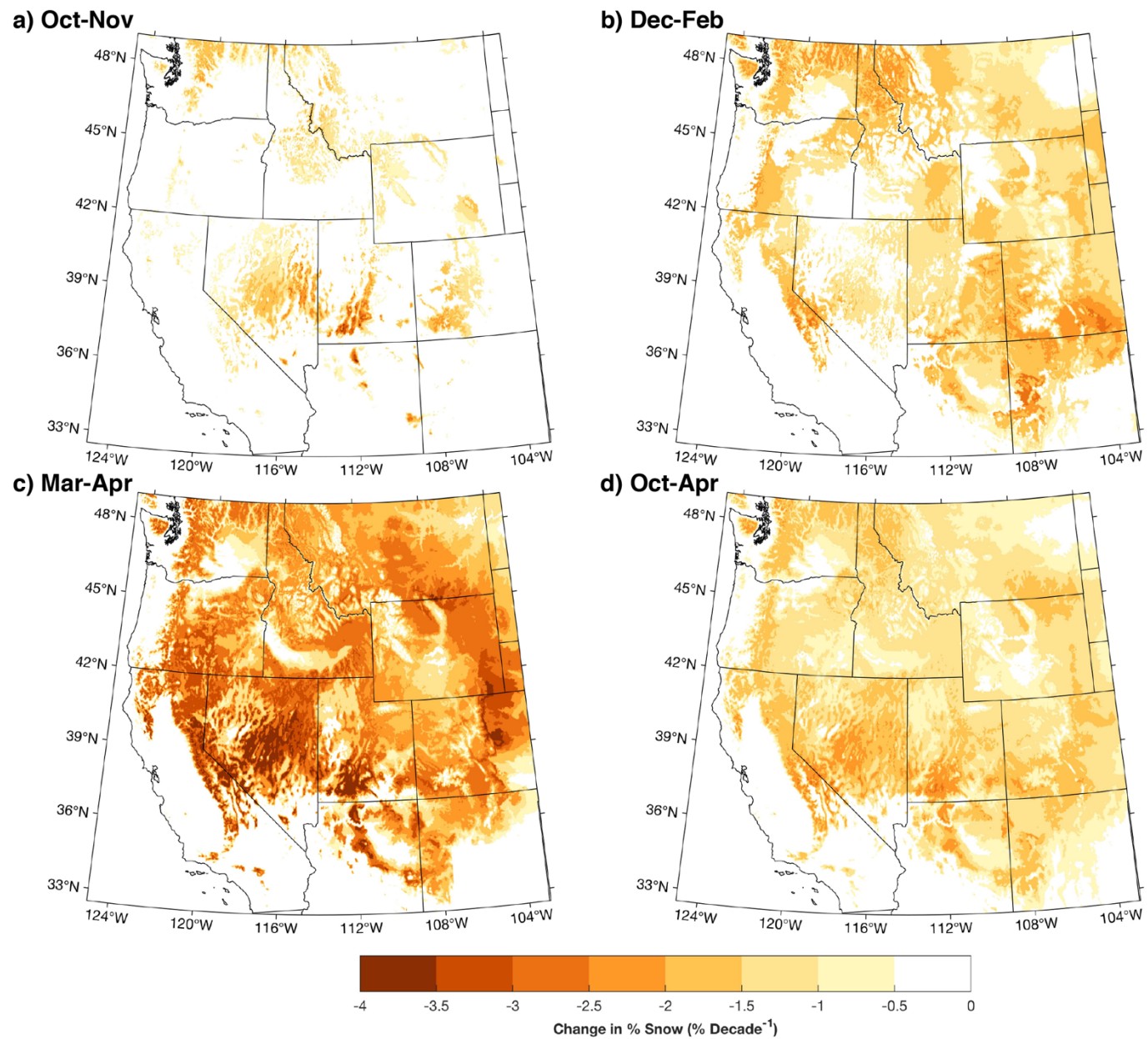

**Figure 6: Decadal trends in %SNOW for the western United States during (a) fall (Oct-Nov), (b) winter (Dec-Feb), (c) spring (Mar-Apr), and (d) for the cool season of the water year (Oct-Apr). Only gridpoints with statistically significant (p<0.05) trends are shown; all gridpoint trends are shown in Supplementary Figure 2.**