# Peer review of "Technical note: Precipitation phase partitioning at landscape-toregional scales"

_Hydrology and Earth System Sciences, 2020_

## Referee Comment (RC1) · Alan M Rhoades (Referee) · 8 May 2020

Summary Lynn et al. in "Technical note: Precipitation phase partitioning at landscape-to-regional scales" unveil a new rain-snow partitioning algorithm, the North American Freezing Level Tracker (NAFLT), and assess trends in California (and western US-wide) snowfall percentages in Winter (Dec-Feb), Spring (Mar-Apr), and Cool Season (Oct-Apr) over the last $\sim$70 years. To build the NAFLT, the authors utilize the NCEP/NCAR reanalysis (2.5-degree resolution) along with the PRISM (4km) reanalysis products. The authors find a more notable decline in rain-snow partitioning in spring (-2/decade to -4%/decade) than winter (-1%/decade to -2%/decade).

Overall, I think the paper by Lynn et al. is well within the scope of the Journal of

[Figure]

Hydrology and Earth System Sciences and a valuable contribution to the scientific community. The figures and results are well-posed and, importantly, the findings have both scientific and societal impact as rain-snow partitioning in mountains (particularly a regular, "healthy" seasonal snowfall total) is a critical assumption in water supply management of western US states.

Most of my comments and revision suggestions are regarding the need to fine-tune the narrative of the manuscript and further discuss/evaluate methodological uncertainties. I would suggest that the editor assign minor revisions to this manuscript.

Review Comments and Suggested Revisions:

Page 1 Line 11 – Change to, "...into rain and snow, particularly snow as it maximizes available water in spring-to-summer."

Line 21 – You might want to cite Huss et al., 2017 here... Huss, M., Bookhagen, B., Huggel, C., Jacobsen, D., Bradley, R., Clague, J., Vuille, M., Buytaert, W., Cayan, D., Greenwood, G., Mark, B., Milner, A., Weingartner, R. and Winder, M. (2017), Toward mountains without permanent snow and ice. Earth's Future, 5: 418-435. doi:10.1002/2016EF000514

Line 23-24 – Change to, "...and, in particular, frozen (snow) components was a foundational assumption of climate stationarity in the development of water management infrastructure and practices..."

Line 35 – Change to, "...Some examples include an upslope shift in winter snow levels..."

Line 37 – What do you mean by "decreased snowpack water storage efficiency"? Does this have to do with cold content decreases and snow ripening occurring more frequently throughout the snow accumulation season? Please clarify.

Page 2 Line 13 – Might want to point to a study (or several) that discuss the dataset/metric inadequacies that water managers/decision makers face when using

climate information. For example... Jagannathan, K., A.D. Jones, and I. Ray, 0: The making of a metric: Co-producing decision-relevant climate science. Bull. Amer. Meteor. Soc., 0, https://doi.org/10.1175/BAMS-D-19-0296.1

Line 19 – Change to, "...scales and, therefore, could be an informative diagnostic for both model development and water resource management in snow dependent regions..."

Line 31 – Change to, "...higher with decreasing latitude where median annual precipitation greatest in the Northern Sierra Nevada..."

Figure 1 caption – Change to, "Estimated historical (1950-1969) percentages of..." In my opinion, the dataset resolution part is TMI in the figure and should just be stated in the methods.

Page 3 Line 3 – Just to clarify, DWR uses the proprietary 800m PRISM product, but did not give you access for this analysis? It would be interesting to know how much of a different answer one would get for rain-snow partitioning if you were to use the 800m vs 4km (i.e., 5x coarsening) PRISM product (particularly in the Southern Cascades)? Similarly, performing a sensitivity analysis of another 5x coarsening (∼20km) of the 4km PRISM product could be informative for climate modelers too. Given that these are diagnostic estimates of rain-snow partitioning, could the authors use the Sierra Nevada Snow Reanalysis (SNSR) from Margulis' group at UCLA - https://margulis-group.github.io/data/ - to explore how different of answer one might get using the author's method vs other methods? This could also include (at least qualitatively) a comparison between more physics-based rain-snow partitioning estimates/trends in the literature versus NAFLT.

Line 20-25 – Might be helpful to cite Jennings et al., 2018 when discussing the "hydrometeor energy balance theory" of snowflakes persisting in above freezing temperatures. Jennings, K.S., Winchell, T.S., Livneh, B. et al. Spatial variation of the rain–snow temperature threshold across the Northern Hemisphere. Nat Commun 9, 1148 (2018).

https://doi.org/10.1038/s41467-018-03629-7 As you expand NAFLT for use beyond the Sierra Nevada (i.e., a more maritime mountain), it might be important to build in (or at least assess the sensitivity of adding in) specific humidity/relative humidity into the rain-snow partitioning algorithm.

Page 4 Line 1-2, Figure 1 – It might be useful to also plot a median snow water year (e.g., 2007-2008)? Also, why not use 1982-1983 for the max snowpack year (DWR's max SWE year - http://cdec.water.ca.gov/snowapp/swcchart.action)?

Line 4-8 - This is beyond the scope of this current study (and seems to be discussed more in Hatchet et al., 2017 and in the "Primary Limitations" section of this article), but given that NCEP/NCAR reanalysis is fairly coarse (2.5-degree resolution) do the authors have a sense of the magnitude of uncertainty baked into rain-snow partitioning estimates in the NAFLT (i.e., confidence intervals)? For example, the freezing isotherm may be influenced by aggregation of sharp gradients in topography in NCEP/NCAR (i.e., resolution dependence) and the precipitation estimates may lack extreme precipitation events (i.e., statistical relationship assumptions in PRISM and/or coarse grid averaging in NCEP/NCAR) and/or may be lower bound estimates of orographic enhancement of storms. The use of the new ECMWF generated ERA5 reanalysis product (i.e., global, 1950-present, hourly/monthly, ~30km, up to ~137 vertical levels) might be a path forward to explore/address any uncertainties in NAFLT too (https://confluence.ecmwf.int/display/CKB/ERA5%3A+data+documentation). At the very least, I think a brief discussion in the manuscript on the potential sources (or even magnitudes and confidence intervals) of uncertainty within the NAFLT rain-snow estimates might be useful and informative to users.

Figure 2 – Is there any value in looking at trends in Oct-Nov too? I am curious if there is an asymmetric or symmetric response in rain-snow partitioning between the "shoulder" months of the Cool Season.

Line 21-30 – Is there any added value in evaluating sliding (rather than fixed) decadal

trend analysis? Or, more specifically (may be a follow-up study), isolate trends based on certain climate variability indices? For example, the ENSO Longitude Index (ELI)... Patricola, C.M., O'Brien, J.P., Risser, M.D. et al. Maximizing ENSO as a source of western US hydroclimate predictability. Clim Dyn 54, 351–372 (2020). https://doi.org/10.1007/s00382-019-05004-8

Line 21-30 - Figure 3 – Do the authors want to discuss potential physical mechanisms regarding the much larger Spring declines in rain-snow partitioning on the leeside (i.e., -4%/decade) compared with windward (i.e., -1-2%\decade) of the Sierra Nevada, particularly in the northern-to-central HUC watersheds? Topography is mentioned but given that there is an asymmetric response between even abutting windward and leeward HUC watersheds (and this is more seen in the Spring rather than Winter), are there potential physical mechanisms that should be discussed? For example, are these changes due to less Spring storms overall or are there the same number of Spring storms, but they are warmer and thus more readily produce rain? Another difference could be that the leeward HUC regions mix trends in the Sierra Nevada with the White Mountains and mask storm-type changes in rain-snow partitioning (e.g., large-scale vs convective and/or inland AR penetration).

Line 28 – Change to, "...remain upslope of the 0 degree C elevation..."

Page 5 Line 5-6, Figure 4 – In addition to watershed area (i.e., proxy for volume of snowpack lost), it might be good to note or discuss other downstream impacts too (i.e., the acre-foot storage of reservoirs, importance of tributaries for surface water, endangered species habitat, etc.). For example, even smaller declines (at least from a water resource management perspective) above Lake Shasta might matter more than more marked declines in watersheds that do not have a reservoir downstream of them (or the reservoir storage capacity is much smaller).

Line 30-31 – Might want to cite a healthy number of future climate modeling studies of the western US here.

Page 6 Line 1-2 – Although a bit tangential to the work in this study, it could be useful to cite some other water supply strategies that can help to offset decreases in mountain snowpack (e.g., recycled water, stormwater catchment, etc.). Some of these supply-side strategies have, historically, been undervalued, but now that co-benefits are being assessed the $/acre-foot start to make more sense and could help to offset the projected low-to-no snow future California might face...

"Economic evaluation of stormwater capture and its multiple benefits in California" - https://journals.plos.org/plosone/article?id=10.1371/journal.pone.0230549

"...current economic analyses of stormwater capture do not adequately examine differences in stormwater project types and do not evaluate co-benefits provided by the projects. As a result, urban stormwater capture is undervalued as a water supply option. To advance economic analyses of stormwater capture, we determined the levelized cost of water in U.S. dollar per acre-foot of water supply (AF; 1 AF = 1233.5 m3) for 50 proposed stormwater capture projects in California, characterizing the projects by water source, process, and water supply yield."

"The cost of alternative urban water supply and efficiency options in California." - https://iopscience.iop.org/article/10.1088/2515-7620/ab22ca

"...this analysis evaluates the costs of four groups of alternatives for urban supply and demand based on data and analysis in the California context: stormwater capture; water recycling and reuse; brackish and seawater desalination; and a range of water conservation and efficiency measures. We also describe some important co-benefits or avoided costs, such as reducing water withdrawals from surface water bodies or polluted runoff in coastal waterways...."

Line 9-10 – I am still on the fence about the argument that "model-based estimates > gridded statistical estimates" for precipitation/snowfall in mountains. There is a lot of nuance that needs to be discussed with this "movement" (which seems primarily "all-in" on WRF). For example, I think some of the assumptions/limitations of micro-

physics/macrophysics schemes and boundary layer schemes in climate models need to be discussed (particularly in the context of mountains). I know this is an on-going debate (and my \$0.02 is one of many), but I would ease the definitive statement regarding "skill" made here.

Line 29 – Change to, "...is that NAFLT can be periodically updated, as datasets become available, with higher resolution gridded data products (citations) and expanded in scope to evaluate global rain-snow partitioning."

---

## Referee Comment (RC2) · Anonymous Referee #2 · 11 May 2020

Review of "Technical note: Precipitation phase partitioning at landscape-to-regional scales" by Lynn et al. Submitted to Hydrology and Earth system sciences. 2020.

General Comments/Overview

This paper describes a new approach developed by the California Department of Water Resources to produce a long term (30+years), monthly, high-resolution (4km), rain/snow partitioning dataset over the Western US. The authors use this dataset/method to estimate long-term changes in rain/snow partitioning. With warmer temperatures, more precipitation is falling as rain rather than snow – which will impact snow water storage and water management practices. The authors argue that due to the paucity of snow observational datasets and the complex topography of the western US multiple datasets are needed to monitor and model hydrologic conditions over

the Western US. Therefore they combine high-resolution PRISM precipitation data with coarse resolution freezing level and fractional snowfall calculations from NCEP/NCAR reanalysis to generate high-resolution fractional snowfall over California (and the Western US). While I believe this is a novel approach and one that has scientific merits, I have deep concerns about the use of the NCEP/NCAR 1 reanalysis product used in this study. In particular, the fact that precipitation from NCEP/NCAR is used to estimate the fraction of precipitation falling as snow. I also do not think the methods used in this paper are adequately described. As this is a technical paper designed to describe a method I believe this paper could be accepted following major revisions.

Major Concerns:

1. The NCEP/NCAR reanalysis dataset used in this study is one of the oldest reanalysis products. At the time of its production/publication the authors (Kalany et al, 1996) state that "C" variables (such as precipitation) are completely determined by the model and should be used with caution. As the fraction of precipitation falling as snow is determined from precipitation in NCEP/NCAR reanalysis, I believe this will add significant uncertainties into the study. At a bare minimum this uncertainty/limitation needs to be discussed in section 5.2 "Primary Limitations" and making sure the reader knows this is a limitation of the study. However I suggest the authors consider performing a similar analysis with a new reanalysis product that adjusts model derived precipitation (e.g. MERRA2 or ERA5) and compare the results with NCEP/NCAR reanalysis.

2. PRISM also provides daily precipitation and surface temperature at 4km resolution. One could estimate daily snowfall using a surface temperature threshold (as in the UA University of Arizona 4km SWE product estimations (https://nsidc.org/data/nsidc-0719; Broxton et al, 2019 and as is done in many land-surface models). The authors need to better explain the science behind why it is more useful or credible to use coarse freezing level data from NCEP/NCAR reanalysis to estimate snowfall percentages, than directly calculating these from PRSIM data. (I don't disagree that this surface temperature is not a great indicator of snow level, its just not explained/justified in the paper).

An interesting comparison could be to look at snowfall estimated from daily T/P vs their method of approximating rain/snow partitioning.

3. The sections describing the NAFLT methods and the DWR approach to rain/snow partitioning are not clear.

a). It seems to me the lowest to the ground freezing level would matter most for snowfall and surface conditions. Why then does the NAFLT method use the uppermost level in areas where there may. be a temperature inversion? Please justify this method.

b) It needs to be made clear that with this method – the freezing level can be below the surface topography.

c) Is the freezing level calculated independently for each 2.5° grid box?

d) Most critical: This statement on Page 3 Line 20 "percent of precipitation that falls at elevations above the 0°C isotherm at 200m increments from 0-4000m" does not make sense to me. Why wouldn't all elevations above the 0° isotherm also be below freezing, and therefore all precipitation would fall as snow? Is there an equation being applied to estimate the fraction of precip falling as snow? Also, does the method start at the freezing level elevation and work up from there (is the reference point for the 0-4000m) then from the freezing level. Or does it start from the surface? Does the method really does is estimate the % of snowfall below the 0° isotherm (where you could have mixed precipitation). Similar language is used in section 3.1 to describe how you apply this method to the high-resolution precip. data from prism (Page 3 Lines 29- 31).

e) In the Primarily Limitations section you state the assumption was made that %snow linearly relates to the NAFTL – but that was not actually stated in the discussion of the methods, it needs to be. My suggestion is to think about how to describe this method to someone who does not know what the freezing level is, or how %snow is calculated and really step through the process – if this is too much detail you might put some of this in supplemental (a diagram could be helpful as well!).

Minor Concerns

- One thing that is missing from the introduction and could be a nice addition is an understanding of how this indicator could actually be used for water resource planning. This is touched upon in the Discussion (Page 5), but I think some of this type of information needs be included in the introduction to further motivate the need for this method. You do say that changes in rain-snow partition are important for water storage – however it wasn't until the Discussion that I could really see why this indicator might actually be used for planning purposes.

- Page 1, Line 23: "components was (replace with: has been) used as a foundation"

- Page 1, Line 27: The use of the word "fate" is a little awkward here, do you mean "phase"?

- Page 1, Line 36: Unclear what you mean by "winter snow levels" here. Do you mean the freezing level in the atmosphere or do you mean increase snow pack (which would be counterintuitive). The jargon of 'winter snow levels" is confusing.

- Page 2, Lines 9-11: The first sentence of this paragraph is not complete. It is unclear what are you incorporating multiple data sources and model outputs into.

- You mention on Page 2, Line 14 that DWR developed a methodology to study historical rain/snow trends at fine spatial resolutions and then on Page 2, Line 16 that the purpose of the note is to provide an updated approach and detail the methods of this indicator. It is unclear to me what in this paper is from the original DWR method and what is the "updated" approach. Is the only difference between the DWR method and the method described in the paper the resolution of the PRISM model data? If the goal of the paper is just to outline the DWR approach – then state that and remove the confusing "updated approach" language. However if the DWR approach is documented elsewhere, and you are documenting changes here in this paper, those differences need to be more explicitly stated.

- Page 2, Line 16: You say "detail the methods of this indicator" but at this point in the paper it is not clear what "indicator" you are talking about. A sentence about the "indicator" before this one is needed. (e.g. Page 6, Line 4 could also be stated here in the text).

- Page 2, Line 19: "and be an important" a modal verb is need before "be" as in "may be" or "can be" etc.

- Page 3, Line 20: ...the percent of precipitation that falls (as snow??) at elevations above the $0°$ isotherm ...

- 2.1 the Study Area Did DWR create this method exclusively for studying trends in rain/snow partitioning, or is this data used in operational forecasts?

- Figure 4 has a number of problems: a). What season is being plotted? Entire water year? Cold season etc? b). You discuss Figure 4b before 4a, they should be flipped in the panel. c). It is unclear from the text how Figure 4b is calculated – what is the IQR and 90%CI based on (is this covering every grid point within that elevation band?). d). Does Figure A represent the values shown in Figure 3 but sorted by elevation?

- Page 5, Line 37 – what is a flood pool? This should be stated in a way that non-flood forecasters/water managers can understand.

- Page 6, Line 4 – this description of the goal of this paper needs to be moved up into the introduction.

Please also note the supplement to this comment:
https://www.hydrol-earth-syst-sci-discuss.net/hess-2020-122/hess-2020-122-RC2-supplement.pdf

---

## Author Response (AR1)

Dear Dr. Viviroli,

Below please find a summary list of changes made to the revised manuscript *"Technical note: Precipitation phase partitioning at landscape-to-regional scales"* by Lynn et al. and submitted to HESS. We addressed all major, minor, and specific reviewer comments (detailed replies are attached) and are pleased to submit a revised manuscript. A few comments requested additional analyses or details that exceed the scope of a technical note intending to highlight a new method, but we included some discussion about these details in the revised manuscript in an effort to motivate continued, and more detailed, research on this topic.

1. Performed an additional analysis where we compared the original method with the older NCEP/NCAR reanalysis to a modern reanalysis product (ERA5). This comparison was a major comment request by both reviewers. Two supplementary figures were added to support our findings of this analysis.
2. We added a panel to Fig. 1 to show another example water year (an average year).
3. We added a panel to Fig. 5 (and Supplementary Fig. 2) to show the Fall trends in the western United States.
4. The panels on Fig. 4 were reversed to match the text.
5. Substantial revisions to the text were made throughout to clarify a number of points raised by reviewers, particularly with respect to the details of the methodology.
6. Where requested, detail was added to improve the description of terminology (e.g., definition of the flood pool).
7. A conceptual diagram was produced and added to the main text with the goal to improve the description of the methodology.
8. Figure captions were corrected and made more descriptive where applicable.
9. Additional references were added where applicable, and we corrected several original references that were initially omitted or incorrect.
10. Addressed all specific comments pertaining to grammar and style.

If you have any additional comments or concerns regarding the manuscript, please do not hesitate to contact me.

Sincerely,
Benjamin Hatchett

Author Responses to Reviewer 1 (Alan Rhoades) for "Technical note: Precipitation phase partitioning at landscape-to-regional scales" by Lynn et al.

Reviewer comments are provided in normal text.
Responses are given in blue
Revised text given in italics (**bold for emphasis**)

Summary Lynn et al. in "Technical note: Precipitation phase partitioning at landscape-to-regional scales" unveil a new rain-snow partitioning algorithm, the North American Freezing Level Tracker (NAFLT), and assess trends in California (and western US-wide) snowfall percentages in Winter (Dec-Feb), Spring (Mar-Apr), and Cool Season (Oct-Apr) over the last~70 years. To build the NAFLT, the authors utilize the NCEP/NCAR reanalysis (2.5-degree resolution) along with the PRISM (4km) reanalysis products. The authors find a more notable decline in rain-snow partitioning in spring (-2/decade to -4%/decade) than winter (-1%/decade to -2%/decade). Overall, I think the paper by Lynn et al. is well within the scope of the Journal of Hydrology and Earth System Sciences and a valuable contribution to the scientific community. The figures and results are well-posed and, importantly, the findings have both scientific and societal impact as rain-snow partitioning in mountains (particularly a regular, "healthy" seasonal snowfall total) is a critical assumption in water supply management of western US states.

Most of my comments and revision suggestions are regarding the need to fine-tune the narrative of the manuscript and further discuss/evaluate methodological uncertainties. I would suggest that the editor assign minor revisions to this manuscript.

We appreciate the positive comments and constructive comments to improve upon the manuscript provided by Dr. Rhoades. Each comment is addressed below.

Review Comments and Suggested Revisions:

Page 1 Line 11 – Change to, "...into rain and snow, particularly snow as it maximizes available water in spring-to-summer."

Thank you for the suggestion. We edited the text following the suggestion, but went for a broader 'warm season' as reservoir deliveries occur from spring through fall:
", particularly snow as it maximizes available water for warm season use"

Line 21 – You might want to cite Huss et al., 2017 here...Huss, M., Bookhagen, B.,Huggel, C., Jacobsen, D., Bradley, R., Clague, J., Vuille, M., Buytaert, W., Cayan,D., Greenwood, G., Mark, B., Milner, A., Weingartner, R. and Winder, M. (2017), Toward mountains without permanent snow and ice. Earth's Future, 5: 418-435.doi:10.1002/2016EF000514

Excellent suggestion, reference has been added.

Line 23-24 – Change to, "...and, in particular, frozen (snow) components was a foundational assumption of climate stationarity in the development of water management infrastructure and practices..."

Thank you for the suggestion to improve this sentence. We made a slight change to the suggested revision to account for the phase partitioning being an assumption as well as the concept of climate stationarity in water management. In other words, precipitation comes as rain and (mostly) snow, and we assume this will not change, so this guides our management strategies.

New text:
"The partitioning of precipitation into liquid (rain) and, ***in particular***, frozen (snow) components ***along with climatic stationarity were foundational assumptions in the development of*** water management infrastructure ***and practices*** in…"

Line 35 – Change to, "...Some examples include an upslope shift in winter snow levels..."

Change made, thank you for the suggested change in phrasing:
"…an ***upslope shift*** in…"

Line 37 – What do you mean by "decreased snowpack water storage efficiency"? Does this have to do with cold content decreases and snow ripening occurring more frequently throughout the snow accumulation season? Please clarify.

Thanks for pointing out our initially confusing text. Your interpretation is valid but not our original intent. We added a brief bit of text better describing the metric used by Das et al. (2009). The ratio of SWE to P declining implies less precipitation is being stored in the snowpack by early spring (e.g., April 1 SWE) and thus the snowpack as a reservoir is less efficient.

New text:
"decreased snowpack water storage efficiency ***as measured by ratios of cool season snow water equivalent to precipitation***"

Page 2 Line 13 – Might want to point to a study (or several) that discuss the dataset/metric inadequacies that water managers/decision makers face when using climate information. For example...Jagannathan, K., A.D. Jones, and I. Ray, 0: The making of a metric: Co-producing decision-relevant climate science. Bull. Amer. Meteor. Soc., 0, https://doi.org/10.1175/BAMS-D-19-0296.1

Great suggestion to include this concept. We added a sentence highlighting this issue:
*"These are among many inadequacies regarding datasets or climate metrics faced by water managers (e.g., Jagannathan et al., 2020)."*

Line 19 – Change to, "...scales and, therefore, could be an informative diagnostic for both model development and water resource management in snow dependent regions..."

Good suggestion to add impact to model development and change 'important' to 'informative'. We made the changes (though we changed the order on model development since the paper is focused on management):
*"We suggest that this approach is scalable to regional-to-continental scales and therefore could be **an informative diagnostic for water resources management and model development in other snowmelt dependent regions.**"*

Line 31 – Change to, "...higher with decreasing latitude where median annual precipitation greatest in the Northern Sierra Nevada..."

Thanks for requesting clarity regarding where the wettest regions are in a latitudinal sense. We re-wrote this sentence as two:
"The elevation distribution of the analysis zones shifts higher with decreasing latitude. ***Median*** annual precipitation is the greatest in the ***higher latitude*** Northern Sierra Nevada and Southern Cascade regions."

Figure 1 caption – Change to, "Estimated historical (1950-1969) percentages of..." In my opinion, the dataset resolution part is TMI in the figure and should just be stated in the methods.

We removed the horizontal resolution part from the caption.
New text:
"*Estimated historical (1950-1969) percentages…*"

Page 3 Line 3 – Just to clarify, DWR uses the proprietary 800m PRISM product, but did not give you access for this analysis?
Yes, DWR uses the 800 m PRISM, and we did initially consider doing the analysis at 800 m. However after discussions, we felt that doing the analysis at the 4 km scale was reasonable from both a physical perspective (see below) but more so since many agencies or groups may not have the resources to pay for the 800 m PRISM products and wanted to show that the method works for the 4 km product.

It would be interesting to know how much of a different answer one would get for rain-snow partitioning if you were to use the 800m vs 4km (i.e., 5x coarsening) PRISM product (particularly in the Southern Cascades)? Similarly, performing a sensitivity analysis of another 5x coarsening (~20km) of the 4km PRISM product could be informative for climate modelers too.
In our preliminary analyses, the results did not appear sensitive to the 800 m vs. 4 km resolution. This is likely because potential differences at finer spatial scales were smoothed out by the elevation-bin size. Spatial differences between the two PRISM products resulting from the

interpolation scheme may also not be physical, since no additional data at finer scales is being included in PRISM (remembering that mountain observations are very sparse to begin with). Further, these spatial differences likely also are canceled out when aggregating to the watershed scales that matter most for water management. We would expect fine scale differences to appear when doing site-specific comparisons (and not aggregating to watershed scales), especially in areas of very complex terrain or large elevation gradients. However, challenges would emerge to test the robustness of these differences in areas where no observations are nearby to ensure that they are physical and not a product of PRISM. This is a limitation with all gridded data products.

We added a note in the limitations section that differences between the PRISM products likely cancel out at the scales of interest here but that site-specific comparisons should show differences:

***"Differences between PRISM products at the 4 km and 800 m scales likely cancel out both from the elevation binning procedure and from the aggregation of data to the watershed scales used by water management. However, we would expect site-specific comparisons to yield differences."***

The coarsening experiment is a good suggestion, and worth investigating further in subsequent work. We added a sentence to the concluding remarks to highlight this:

"The main advantage of the described approach is that the NAFLT can be periodically updated as higher resolution gridded data products become available (e.g., TerraClimate; Abatzoglou et al., 2018). ***It could also be expanded in scope to evaluate global rain-snow partitioning in global or regional climate models by aggregating to the spatial resolutions used in these models.***"

Given that these are diagnostic estimates of rain-snow partitioning, could the authors use the Sierra Nevada Snow Reanalysis (SNSR) from Margulis' group at UCLA - https://margulis-group.github.io/data/ - to explore how different of answer one might get using the author's method vs other methods? This could also include (at least qualitatively) a comparison between more physics-based rain-snow partitioning estimates/trends in the literature versus NAFLT.

This is a great suggestion, and something we are actively working on. One limitation is the robustness of the SNSR at elevations below 1500 m: "The reanalysis dataset presented herein covers 20 watersheds and is applied to elevations above 1500 m, which represents the nominal snow line (Bales et al. 2006; Guan et al. 2013)" (quoted from Margulis et al. 2016). While beyond the scope of this study, as this is intended as a technical note to describe a general methodology with the hope/intent to inspire work exactly as the reviewer noted, we are also exploring other SWE reanalyses and remote sensing products. Our approach does not technically resolve SWE, but rather snowfall liquid water equivalent. Hence comparisons with SWE products would be flawed by not considering ablation processes. That all said, we added a line to the concluding remarks section describing how snow reanalyses offer a complementary approach to other methods of analyzing changes in mountain snowpack:

Added sentence:
*"These products provide complementary information to high resolution snow reanalyses that incorporate satellite and/or in situ data (e.g., Margulis et al., 2016; Zeng et al. 2018)."*

Added citations:
Margulis, S. A., Cortés, G., Girotto, M., and Durand, M.: A Landsat-era Sierra Nevada snow reanalysis (1985–2015), J. Hydrometeor, 17(4), 1203-1221, 2016.

Zeng, X., Broxton, P., and Dawson, N.: Snowpack change from 1982 to 2016 over conterminous United States. Geophys. Res. Lett*.*, 45, 12,940– 12,947. https://doi.org/10.1029/2018GL079621, 2018.

Line 20-25 – Might be helpful to cite Jennings et al., 2018 when discussing the "hydrometeor energy balance theory" of snowflakes persisting in above freezing temperatures. Jennings, K.S., Winchell, T.S., Livneh, B. et al. Spatial variation of the rain–snow temperature threshold across the Northern Hemisphere. Nat Commun 9, 1148 (2018) https://doi.org/10.1038/s41467-018-03629-7 As you expand NAFLT for use beyond the Sierra Nevada (i.e., a more maritime mountain), it might be important to build in (or at least assess the sensitivity of adding in) specific humidity/relative humidity into the rain-snow partitioning algorithm.

This is a great suggestion, we added the citation and also added a line to the limitations section about including RH or wet bulb temperature (among other variables) as potential ways to further improve the method:
*"Further, comparisons with approaches that include relative humidity or wet bulb temperatures are recommended to further improve the methodology, as these have been shown to improve the quality of rain-snow partitioning (Harpold et al., 2017, Wang et al., 2019)."*

*Added citations:*
Harpold, A. A., Rajagopal, S., Crews, J. B., Winchell, T., and Schumer, R.: Relative humidity has uneven effects on shifts from snow to rain over the western US, Geophys. Res. Lett., 44(19), 9742-9750, doi:10.1002/2017GL075046, 2017.

Wang, Y. -H., Broxton, P., Fang, Y., Behrangi, A., Barlage, M., Zeng, X., and Niu, G. -Y.: A wet-bulb temperature-based rain-snow partitioning scheme improves snowpack prediction over the drier Western United States, Geophys. Res. Lett., 46, 13825– 13835, https://doi.org/10.1029/2019GL085722, 2019.

Page 4 Line 1-2, Figure 1 – It might be useful to also plot a median snow water year(e.g., 2007-2008)? Also, why not use 1982-1983 for the max snowpack year (DWR's max SWE year - http://cdec.water.ca.gov/snowapp/swcchart.action)?

We see the reviewers point, and have changed the lower panels of Figure 1 to better show examples of interannual variability. We used the suggestion for 2008 as the median year (b) and have a low %$_{SNOW}$ year (2015; panel (a)) and a high %$_{SNOW}$ year (1980; panel (c)). Our new Figure 1 is as follows:

[Figure]

"Figure 1: Estimated historical (1950-1969) percentages of precipitation as snow for (a) winter (Dec-Feb), (b) spring (Mar-Apr), and (c) for the full cool season (Oct-Apr). Examples of %$_{SNOW}$ averaged over the cool season (October-April) of water years (d) 2015, *(e) 2008*, and (f) 1980. Thick black contours denote California Department of Water Resources analysis zones."

To address the reviewer's point, we did generate a plot of WY1983. However, it appears less snowy than 1980. This is likely a result of the signal of several warmer-than-normal storms during 1983 (recall there were some substantial flood events) and provides an example showing how %$_{SNOW}$ and SWE are not always directly linked. If one is measuring in terms of SWE, additional water added to the snowpack through rain (and under the assumption that this water was stored in the snowpack) could result in a bigger SWE year than a year that had all snow but less overall precipitation.

[Figure]

Fraction of precipitation as snow during water year 1983 (left) versus water year 1980 (right).

Line 4-8 - This is beyond the scope of this current study (and seems to be discussed more in Hatchet et al., 2017 and in the "Primary Limitations" section of this article), but given that NCEP/NCAR reanalysis is fairly coarse (2.5-degree resolution) do the authors have a sense of the magnitude of uncertainty baked into rain-snow partitioning estimates in the NAFLT (i.e., confidence intervals)? For example, the freezing isotherm may be influenced by aggregation of sharp gradients in topography in NCEP/NCAR (i.e., resolution dependence) and the precipitation estimates may lack extreme precipitation events (i.e., statistical relationship assumptions in PRISM and/or coarse grid averaging in NCEP/NCAR) and/or may be lower bound estimates of orographic enhancement of storms. The use of the new ECMWF generated ERA5 reanalysis product (i.e., global, 1950-present, hourly/monthly,~30km, up to~137 vertical levels) might be a path forward to explore/address any uncertainties in NAFLT too (https://confluence.ecmwf.int/display/CKB/ERA5%3A+data+documentation). At the very least, I think a brief discussion in the manuscript on the potential sources (or even magnitudes and confidence intervals) of uncertainty within the NAFLT rain-snow estimates might be useful and informative to users.

These are excellent points and similar concerns with NCEP/NCAR were also brought up by the other reviewer. Following both reviewer's suggestions, we repeated the analysis with ERA-5 for the four aggregated DWR watersheds to provide some estimates of how well NCEP/NCAR performs. We found encouraging results (figure below, added as a supplementary figure), with ERA-5 and NCEP/NCAR being very well-correlated over the overlapping time period (correlations exceeding 0.9). ERA-5 was a bit colder (more %$_{SNOW}$), which is likely related to a number of improvements in the ERA-5 model compared to NCEP/NCAR (data assimilation, spatial/vertical resolution, terrain, physical process representation). We added a paragraph to the limitations section highlighting our use of an older model (which was state-of-the-art at the time the NAFLT was developed in ~2008) and showing that it still performs relatively well. All in all, this comparison suggests that the method we are showing is valid and can be a way to evaluate precipitation partitioning in models.

*"The NCEP/NCAR reanalysis, which the NAFLT uses to identify freezing levels and partition precipitation, is an older generation reanalyses product. Recent advances in atmospheric reanalyses such as ERA-5 (Hersbach et al., 2020) provide advances in data assimilation procedures, have finer spatiotemporal resolution, and provide 0°C heights as standard products. A comparison of the NCEP/NCAR approach to ERA-5 during 1979-2018 showed strong similarity in the spatial distribution of %$_{SNOW}$ (Supplementary Figure 3) and high interannual correlations (0.9<R<0.99), with slightly higher %$_{SNOW}$ in ERA-5 (Supplementary Figure 4). The method for partitioning precipitation described herein shows promise using the older NCEP/NCAR reanalysis, but it flexible enough to incorporate advances in reanalyses products as well as climate model projections."*

New Supplementary Figures have been added to the revised manuscript:

[Figure]

[Figure]

**Fraction of Precipitation Falling as SNOW**
**(x 100 = %$_{SNOW}$)**

*Supplementary Figure 3: Comparison of 1981-2010 mean water year fraction of precipitation falling as snow (multiply by 100 to yield %$_{SNOW}$) for northern California and western Nevada produced using ERA-5 (left) with NCEP-NCAR (right).*

[Figure]

*Supplementary Figure 4: Comparison of fraction of precipitation falling as snow for ERA-5 (blue line) and NCEP-NCAR (red line) for the period 1979-2018 for the four DWR analysis zones, ordered clockwise from upper left: Southern Cascades, Northern Sierra Nevada, Central Sierra Nevada, and Southern Sierra Nevada.*

Figure 2 – Is there any value in looking at trends in Oct-Nov too? I am curious if there is an asymmetric or symmetric response in rain-snow partitioning between the "shoulder" months of the Cool Season.

Fall trends were not nearly as strong in California as other seasons, and west-wide there were only a few locations of stronger signals (leeside of the WA Cascades, central Great Basin, southern Utah, higher elevations in the Rockies) so initially we omitted these results.

Looking more closely, these trends are interesting since they do affect the highest elevations (CO Rockies, Wind Rivers, NW Montana ranges). There are also interesting signals in the eastern Great Basin, southern Utah, northern Arizona, and the northern Cascades of Washington. We will leave the main manuscript figures showing California as they are, but now include fall in the west-wide Figure 5:

New Figure 5:

[Figure]

**a) Oct-Nov**

**b) Dec-Feb**

**c) Mar-Apr**

**d) Oct-Apr**

Change in % Snow (% Decade$^{-1}$)

Line 21-30 – Is there any added value in evaluating sliding (rather than fixed) decadal trend analysis? Or, more specifically (may be a follow-up study), isolate trends based on certain climate variability indices? For example, the ENSO Longitude Index (ELI)...Patricola, C.M., O'Brien, J.P., Risser, M.D. et al. Maximizing ENSO as a source of western US hydroclimate predictability. Clim Dyn 54, 351–372 (2020). https://doi.org/10.1007/s00382-019-05004-8 This is a great follow-up study suggestion, and the exact direct we'd like to go (in addition to improving the calculation of the metric). For example, Abatzoglou 2011 did find that trends in the PNA had contributed to a hastening of freezing level increases and declines in precipitation as snow; additional exploration of how modes of variability influence freezing levels would certainly add value.

While beyond the scope of this methods paper to evaluate modes of variability, we have added a note that this would be a fruitful area of further research:

*"Further examination of how freezing levels are influenced by large scale modes of climate variability are also recommended. For example, Abatzoglou (2011) found trends in the Pacific-*

*North American pattern contributed to increases in freezing levels and declines in precipitation falling as snow. Evaluating freezing level and precipitation phase relationships to isolated modes of climate variability may provide useful guidance for hydroclimate predictability at lead times relevant for water management (e.g., Patricola et al. 2020)."*

Line 21-30 – Figure 3 – Do the authors want to discuss potential physical mechanisms regarding the much larger Spring declines in rain-snow partitioning on the leeside (i.e.,-4%/decade) compared with windward (i.e., -1-2%\decade) of the Sierra Nevada, particularly in the northern-to-central HUC watersheds? Topography is mentioned but given that there is an asymmetric response between even abutting windward and leeward HUC watersheds (and this is more seen in the Spring rather than Winter), are there potential physical mechanisms that should be discussed? For example, are these changes due to less Spring storms overall or are there the same number of Spring storms, but they are warmer and thus more readily produce rain? Another difference could be that the leeward HUC regions mix trends in the Sierra Nevada with the White Mountains and mask storm-type changes in rain-snow partitioning (e.g., large-scale vs convective and/or inland AR penetration).

We appreciate the suggestion to add some discussion on the windward/leeward and spring trends. There are likely dynamic explanations for these trends, however without substantial effort that goes beyond the scope of a methods paper, we would be left speculating. We have included additional text that the method described can help identify curious spatial behaviors that warrant additional research to provide a physical explanation:
*"The apparent asymmetric warming of the leeside of the Sierra Nevada compared to the windward side (Fig. 2) warrants additional investigation to elucidate physical mechanisms generating this asymmetry. The watershed-scale signal may also be a by-product of the greater land area at higher elevation in leeside watersheds. A benefit of the spatially distributed nature of the DWR approach is that it facilitates the identification of spatial behaviours that may not be readily apparent in station observations."*

Line 28 – Change to, "...remain upslope of the 0 degree C elevation..."
Change made, thank you for the suggestion.

Page 5 Line 5-6, Figure 4 – In addition to watershed area (i.e., proxy for volume of snowpack lost), it might be good to note or discuss other downstream impacts too (i.e., the acre-foot storage of reservoirs, importance of tributaries for surface water, endangered species habitat, etc.). For example, even smaller declines (at least from a water resource management perspective) above Lake Shasta might matter more than more marked declines in watersheds that do not have a reservoir downstream of them (or the reservoir storage capacity is much smaller).
Thank you for bringing up the need to include these discussion points. We added a sentence to briefly point out these impacts, as our metric could be much more (or less) useful for basins that are more (or less) susceptible to precipitation phase changes.

New text to get the idea in there:

*"In watersheds with minimal or no reservoir storage, changes from snow to rain may have more impactful changes on flood hazard and habitat, especially during warm season low flows, thus requiring more creative or costly solutions."*

Line 30-31 – Might want to cite a healthy number of future climate modeling studies of the western US here.

Good suggestion, we added several studies to this sentence:
*"...(Klos et al., 2014; Huang et al., 2018; Rhoades et al., 2018a; Sun et al., 2018)."*

We also added a sentence further up to better connect with other Sierra Nevada-specific modeling and projection studies:

**"Snowpack declines are robustly projected to continue into the 21st century (Rhoades et al., 2018a) and be further exacerbated during droughts (Berg and Hall, 2017) and extreme wet years (Huang et al., 2018)."**

*Added citations:*

Berg, N., and Hall, A.: Anthropogenic warming impacts on California snowpack during drought, Geophys. Res. Lett., 44, 2511– 2518, doi:10.1002/2016GL072104, 2017.

Huang, X., Hall, A. D., and Berg, N.: Anthropogenic warming impacts on today's Sierra Nevada snowpack and flood risk. Geophys Res Lett, 45, 6215– 6222, https://doi.org/10.1029/2018GL077432, 2018.

Rhoades, A. M., Ullrich, P. A., & Zarzycki, C. M. Projecting 21st century snowpack trends in Western USA mountains using variable-resolution CESM. Clim. Dyn., 50(1), 261– 288. https://doi.org/10.1007/s00382-017-3606-0, 2018b.

Sun, F., Berg, N., Hall, A., Schwartz, M., and Walton, D.: Understanding end-of-century snowpack changes over California's Sierra Nevada. Geophys. Res. Lett., 46, 933– 943. https://doi.org/10.1029/2018GL080362, 2019.

Page 6 Line 1-2 – Although a bit tangential to the work in this study, it could be useful to cite some other water supply strategies that can help to offset decreases in mountain snowpack (e.g., recycled water, stormwater catchment, etc.). Some of these supply-side strategies have, historically, been undervalued, but now that co-benefits are being assessed the $/acre-foot start to make more sense and could help to offset the projected low-to-no snow future California might face..."Economic evaluation of stormwater capture and its multiple benefits in California" - https://journals.plos.org/plosone/article?id=10.1371/journal.pone.0230549"...current economic analyses of storm water capture do not adequately examine differences in stormwater project types and do not evaluate co-benefits provided by the projects. As a result, urban stormwater capture is undervalued as a water supply option. To advance economic analyses of stormwater capture, we determined the levelized cost of water in U.S. dollar per acre-foot of water supply

(AF; 1 AF = 1233.5 m3) for 50 proposed stormwater capture projects in California, characterizing the projects by water source, process, and water supply yield." "The cost of alternative urban water supply and efficiency options in California." - https://iopscience.iop.org/article/10.1088/2515-7620/ab22ca"...this analysis evaluates the costs of four groups of alternatives for urban supply and demand based on data and analysis in the California context: stormwater capture; water recycling and reuse; brackish and seawater desalination; and a range of water conservation and efficiency measures. We also describe some important co-benefits or avoided costs, such as reducing water withdrawals from surface water bodies or polluted runoff in coastal waterways...."

We appreciate the suggestions to dive a little deeper into this and have added to the discussion section (in **bold italics**) that already gained additional insight from a previous reviewer comment (*italics*):
"*In watersheds with minimal or no reservoir storage, changes from snow to rain may have more impactful changes on flood hazard and habitat, especially during low warm season flows, thus requiring more creative or costly solutions. **Other non-traditional strategies to offset projected decreases in mountain snowpack and achieve water supply reliability exist, such as storm water recapture, water recycling, and water markets. However, these will require economic assessments to determine feasibility (Cooley et al., 2019).***"

Added reference:
*Cooley, H., Phurisamban, R. and Gleick, P., 2019. The cost of alternative urban water supply and efficiency options in California. Environmental Research Communications, 1(4), p.042001.*

Line 9-10 – I am still on the fence about the argument that "model-based estimates> gridded statistical estimates" for precipitation/snowfall in mountains. There is a lot of nuance that needs to be discussed with this "movement" (which seems primarily "all-in" on WRF). For example, I think some of the assumptions/limitations of micro-physics/macrophysics schemes and boundary layer schemes in climate models need to be discussed (particularly in the context of mountains). I know this is an on-going debate (and my $0.02 is one of many), but I would ease the definitive statement regarding "skill" made here.

Agreed, we revised this sentence to ease the definition about skill and be more qualitative ("more realistic").

New text:
"*Indeed, some high-resolution model simulations show more realistic precipitation amounts in mountains than some observational networks (Lundquist et al., 2020; Wrzesien et al., 2019).*"

Line 29 – Change to, "...is that NAFLT can be periodically updated, as datasets become available, with higher resolution gridded data products (citations) and expanded in scope to evaluate global rain-snow partitioning."

Thank you for the suggestion, we revised the text as follows:

"The main advantage of the described approach is that ***the NAFLT can be periodically updated as higher resolution gridded data products become available, including those at global scales*** (e.g., TerraClimate; Abatzoglou et al., 2018) ***and global and regional climate models***."

Responses to Reviewer 2 for "Technical note: Precipitation phase partitioning at landscape-to-regional scales" by Lynn et al.

Reviewer comments are provided in normal text.
Responses are given in blue
Revised text given in italics (**bold for emphasis**)

Review of "Technical note: Precipitation phase partitioning at landscape-to-regional scales" by Lynn et al. Submitted to Hydrology and Earth system sciences. 2020. General Comments/Overview This paper describes a new approach developed by the California Department of Water Resources to produce a long term (30+years), monthly, high-resolution (4km), rain/snow partitioning dataset over the Western US. The authors use this dataset/method to estimate long-term changes in rain/snow partitioning. With warmer temperatures, more precipitation is falling as rain rather than snow –which will impact snow water storage and water management practices. The authors argue that due to the paucity of snow observational datasets and the complex topography of the western US multiple datasets are needed to monitor and model hydrologic conditions over the Western US. Therefore they combine high-resolution PRISM precipitation data with coarse resolution freezing level and fractional snowfall calculations from NCEP/NCAR reanalysis to generate high-resolution fractional snowfall over California (and the Western US). While I believe this is a novel approach and one that has scientific merits, I have deep concerns about the use of the NCEP/NCAR 1 reanalysis product used in this study. In particular, the fact that precipitation from NCEP/NCAR is used to estimate the fraction of precipitation falling as snow. I also do not think the methods used in this paper are adequately described. As this is a technical paper designed to describe a method I believe this paper could be accepted following major revisions.

We appreciate the constructive comments provided by the reviewer and have majorly revised the paper in order to address their major, minor, and specific concerns.

Major Concerns:
1. The NCEP/NCAR reanalysis dataset used in this study is one of the oldest reanalysis products. At the time of its production/publication the authors (Kalany et al, 1996) state that "C" variables (such as precipitation) are completely determined by the model and should be used with caution. As the fraction of precipitation falling as snow is determined from precipitation in NCEP/NCAR reanalysis, I believe this will add significant uncertainties into the study. At a bare minimum this uncertainty/limitation needs to be discussed in section 5.2 "Primary Limitations" and making sure the reader knows this is a limitation of the study. However I suggest the authors consider performing a similar analysis with a new reanalysis product that adjusts model derived precipitation (e.g. MERRA2 or ERA5) and compare the results with NCEP/NCAR reanalysis.

We appreciate the reviewer's concerns about the NCEP/NCAR reanalysis and thank them for requesting additional information regarding the uncertainties as well as the suggestion to perform

a similar analysis with a modern reanalysis. Similar concerns with NCEP/NCAR were also brought up by Reviewer 1.

Following both reviewer's suggestions, we repeated the analysis with ERA-5 for the four aggregated DWR watersheds to provide some estimates of how well NCEP/NCAR performs. We found encouraging results (figures below, added as supplementary figures), with ERA-5 and NCEP/NCAR being very well-correlated over the overlapping time period (correlations exceeding 0.9). ERA-5 was a bit colder (more %SNOW), which is likely related to a number of improvements in the ERA-5 model compared to NCEP/NCAR (data assimilation, spatial/vertical resolution, terrain, physical process representation). We added a paragraph to the limitations section highlighting our use of an older model (which was state-of-the-art at the time the NAFLT was developed in ~2008) and showing that it still performs relatively well. All in all, this comparison suggests that the method we are showing is valid (despite limitations in NCEP/NCAR) and thus the method represents a useful way to evaluate precipitation partitioning in models and distribute this partitioning across landscapes when linked with a gridded precipitation product such as PRISM.

Added text:

*"The NCEP/NCAR reanalysis, which the NAFLT uses to identify freezing levels and partition precipitation, is an older generation reanalyses product. Recent advances in atmospheric reanalyses such as ERA-5 (Hersbach et al., 2020) provide advances in data assimilation procedures, have finer spatiotemporal resolution, and provide 0°C heights as standard products. A comparison of the NCEP/NCAR approach to ERA-5 during 1979-2018 showed strong similarity in the spatial distribution of %$_{SNOW}$ (Supplementary Figure 3) and high interannual correlations (0.9<R<0.99), with slightly higher %$_{SNOW}$ in ERA-5 (Supplementary Figure 4). The method for partitioning precipitation described herein shows promise using the older NCEP/NCAR reanalysis, but it flexible enough to incorporate advances in reanalyses products as well as climate model projections."*

New Supplementary Figures were added to the revised manuscript:

**a) ERA5 1981-2010 mean**

[Figure]

**b) NCEP/NCAR 1981-2010 mean**

[Figure]

0.1 0.2 0.3 0.4 0.5 0.6 0.7 0.8 0.9
**Fraction of Precipitation Falling as SNOW**
**(x 100 = %SNOW)**

0.1 0.2 0.3 0.4 0.5 0.6 0.7 0.8 0.9
**Fraction of Precipitation Falling as SNOW**
**(x 100 = %SNOW)**

*Supplementary Figure 3: Comparison of 1981-2010 mean water year fraction of precipitation falling as snow (multiply by 100 to yield %SNOW) for northern California and western Nevada produced using ERA-5 (left) with NCEP-NCAR (right).*

[Figure]

*Supplementary Figure 4: Comparison of fraction of precipitation falling as snow for ERA-5 (blue line) and NCEP-NCAR (red line) for the period 1979-2018 for the four DWR analysis zones, ordered clockwise from upper left: Southern Cascades, Northern Sierra Nevada, Central Sierra Nevada, and Southern Sierra Nevada.*

2.PRISM also provides daily precipitation and surface temperature at 4km resolution. One could estimate daily snowfall using a surface temperature threshold (as in the UA University of Arizona 4km SWE product estimations (https://nsidc.org/data/nsidc-0719; Broxton et al, 2019 and as is done in many land-surface models). The authors need to better explain the science behind why it is more useful or credible to use coarse freezing level data from NCEP/NCAR reanalysis (or another reanalysis product in 1.) to estimate snowfall percentages, than directly calculating these from PRSIM data. (I don't disagree that this surface temperature is not a great indicator of snow level, its just not explained/justified in the paper). An interesting comparison could be to look at snowfall estimated from daily T/P vs their method of approximating rain/snow partitioning.

We appreciate the reviewer's request for better justification about the use of freezing level data instead of surface-based temperatures. There are a number of issues with surface temperatures (data sparseness especially in complex terrain, inadequacies in resolving near-surface lapse rates (e.g., Lute and Abatzoglou, 2020)). These errors can be translated into significant errors in snow

models. Another issue with daily-based gridded products are the daily time step that may miss the dynamics of change in snow level throughout the day (such as abrupt rises or falls with frontal passage). The 6-hourly approach with reanalysis may help address this issue somewhat. As a synoptic scale phenomenon, the freezing level is generally well-resolved by models (admittedly there can be substantial variation at finer scales due to microphysics/latent heating/hydrometeor dragging). We added additional text to the introduction to further motivate the study and the use of freezing elevations. We will continue thinking about this issue as we revise the paper, as this is a very important consideration to incorporate into the manuscript well.

New text:
"*Daily gridded products based on sparse observational networks in mountainous areas have their own suite of limitations, such as capturing sub-daily fluctuations in temperature or resolving lapse rates (Lute and Abatzoglou 2020).*"

"The purpose of this technical note is to describe the development of this diagnostic indicator aimed at quantifying how rain and snow are partitioned **based upon the elevation of the atmospheric freezing (0°C) isotherm, which has been found to be well-resolved by global models in complex terrain (Abatzoglou 2011).**"

Added reference:
Lute, AC, Abatzoglou, JT. Best practices for estimating near-surface air temperature lapse rates. *Int J Climatol*. 2020; 1– 16. https://doi.org/10.1002/joc.6668

We agree it would be interesting to compare this approach to a surface-based approach and appreciate the suggestion. While beyond the scope of this methods paper to dive into comprehensive comparisons, we have added text to encourage this type of study using surface-based data (especially if the surface data includes RH or other necessary parameters to calculate wet bulb temperatures):

"**Further, comparisons with approaches that include relative humidity or wet bulb temperatures are recommended to further improve the methodology, as these have been shown to improve the quality of rain-snow partitioning (Harpold et al., 2017, Wang et al., 2019).**"

In the concluding remarks, we note our method is complementary to other approaches (such as snow reanalyses), and hope to encourage more detailed comparisons between the suite of available products:

"*These products provide complementary information to high resolution snow reanalyses that incorporate satellite and/or in situ data (e.g., Margulis et al., 2016; Zeng et al. 2018).*"

3. The sections describing the NAFLT methods and the DWR approach to rain/snow partitioning are not clear.

We appreciate the request for improving our description, and provide specific responses below.

It seems to me the lowest to the ground freezing level would matter most for snowfall and surface conditions. Why then does the NAFLT method use the uppermost level in areas where there may be a temperature inversion? Please justify this method.

Perhaps surprisingly, the surface temperature (what I am interpreting by 'ground freezing level'), does not have the 'final say' in precipitation phase. A multitude of physical processes, such as latent heating, hydrometeor fall speed, and others control the melting of a snowflake (see Minder et al., 2011 and Jennings et al., 2018 for further descriptions of processes; note the other reviewer suggested the additional Jennings reference). This means that the freezing level elevation is a maximum estimate of where snow may turn to rain, but it is often below (typically 100-300 meters) that elevation, hence why snow can be experienced at surface temperatures above freezing. Further, soundings may indicate freezing rain in overrunning types of situations (cold air pooling at the surface and being overrun by warmer air; this would produce an inversion in a sounding), which is not snow. We are avoiding this situation by following standard NWP definitions of freezing level. We approached this issue in the discussion, but we agree with the reviewer that some up-front justification would help readers when introducing the method.

With regards to the inversion issue, our original text used the standard NWP calculation (also used in ERA-5 but we slightly revised it for clarity: "The uppermost atmospheric level below which the 0°C isotherm occurs is considered for cases in which the vertical temperature profile includes inversion conditions *with* multiple incursions of the 0°C isotherm"

We have revised the text to improve the clarity of the description of the method and provide additional justification:
"*The* NAFLT calculates the freezing level as the **highest elevation in the troposphere (200-1000 hPa)** above mean sea level where **free-air temperatures are 0°C.** If the entire atmosphere is at or below freezing on a given 6-hr period, a value of zero meters above mean sea level is provided. The uppermost atmospheric level below which the 0°C isotherm occurs is considered for cases in which the vertical temperature profile includes inversion conditions **with** multiple incursions of the 0°C isotherm. In addition to providing estimates of the elevation of the 0°C isotherm, the NAFLT calculates the **monthly** percent of precipitation that falls **as snow (%SNOW) at 200 m elevational increments from 0-4000 m. This is done by assigning all 6-hourly modelled precipitation from the NCEP/NCAR reanalysis as snow for elevations above the corresponding freezing level and all precipitation in a 6-hour increment as rain for elevations below the freezing level.** The freezing level is a very conservative estimate of the snow level as precipitation can often persist as snow below the freezing elevation due to latent heat fluxes (e.g., snow falling in a sub-saturated atmosphere, deep isothermal temperature profiles, or during heavy precipitation episodes that entrains colder air to lower levels in the atmosphere; **Minder et al., 2011; Jennings et al., 2018**). However, accumulations of snow below the elevation of the 0°C isotherm may be transient due to nominal cold content of snow."

We also revised the DWR approach text in section 3.1:
*The DWR approach calculates % $_{SNOW}$ by first bilinearly interpolating of %$_{SNOW}$ horizontally for each 200 m elevational increment from NAFLT and then assigning %$_{SNOW}$ to each fine-scale grid point per the smallest elevational difference between fine-scale elevation (e.g., 4-km DEM) and*

*the 200 m elevational bins. If the freezing level elevation is below the terrain elevation, all precipitation falls as snow (%SNOW = 100). Given the known inadequacies of coarse-scale reanalysis precipitation fields in complex terrain, we multiplied estimates of monthly PRISM precipitation by %SNOW to partition precipitation between total frozen (%SNOW) and liquid (%RAIN) components similar to Abatzoglou (2011).*

It needs to be made clear that with this method –the freezing level can be below the surface topography.

Thank you for the suggestion, we added a sentence to make this clear:
*"If the freezing level is below the terrain, all precipitation falls as snow (%SNOW = 100)."*

Is the freezing level calculated independently for each 2.5º grid box?

Correct, we revised and added text to reflect this:
"*free-air temperatures are 0°C **for each 2.5° NCEP/NCAR grid point**"*

"*The DWR approach calculates % SNOW by first bilinearly interpolating the 2.5° grid point estimates of %SNOW horizontally for each 200 m elevational increment from the NAFLT. Then it assigns %SNOW to each fine-scale PRISM grid point per the smallest elevational difference between fine-scale elevation (e.g., 4 km DEM) and the 200 m elevational bins."*

Most critical: This statement on Page 3 Line 20 "percent of precipitation that falls at elevations above the 0ºC isotherm at 200m increments from 0-4000m" does not make sense to me. Why wouldn't all elevations above the 0º isotherm also be below freezing, and therefore all precipitation would fall as snow? Is there an equation being applied to estimate the fraction of precip falling as snow?

Thank you for pointing out the confusing nature of this sentence in its original form. This sentence has been revised for clarity:
"*This is done by assigning all 6-hourly modeled precipitation from NCEP/NCAR **reanalysis as snow for elevations above the corresponding freezing level**"*

Also, does the method start at the freezing level elevation and work up from there (is the reference point for the 0-4000m) then from the freezing level. Or does it start from the surface? Does the method really estimate the % of snowfall below the 0º isotherm (where you could have mixed precipitation). Similar language is used in section 3.1 to describe how you apply this method to the high-resolution precip data from prism (Page 3 Lines 29-31).

Per standard approaches with Numerical Weather Prediction models and freezing-level outputs that are now part of modern reanalyses (e.g., ERA-5), the 0°C level is defined as the highest atmospheric level in the troposphere where the 0°C level is crossed. This is done to avoid

inversions in the case of freezing rain/sleet that might otherwise assign the precipitation type as snow.  We further clarified that the approach used is binary in the sense that for a given 6-hour period, elevations above the freezing level are assigned 100% snow, and those below 0% snow. Please see major revisions to the text provided above.

In the Primarily Limitations section you state the assumption was made that %snow linearly relates to the NAFTL –but that was not actually stated in the discussion of the methods, it needs to be. My suggestion is to think about how to describe this method to someone who does not know what the freezing level is, or how %snow is calculated and really step through the process– if this is too much detail you might put some of this in supplemental (a diagram could be helpful as well!).

We appreciate the suggestion to describe the method in a more step-by-step manner. Please see specific changes noted above and in the revised manuscript that includes a heavily-revised methods section. Including a diagram is an excellent idea to visually explain the method, and we thank the reviewer for this suggestion. We now include a schematic in the revised manuscript (see next page) and have referenced the diagram throughout the sections describing the NAFLT and the DWR approach.

**Step 1:** Working downward from 200 hPa, identify height of freezing level for each NCEP/NCAR grid cell at each 6 hr time step via linear interpolation, assign to 200 m bin (from 0 - 4000 m)

[Figure]

**Step 2:** Assign phase to precipitation based on freezing level for bins above and below.

[Figure]

**Step 3:** For each 200 m elevation bin at each grid point, sum Psnow and Prain over all time steps in the month then divide by total P to calculate monthly %SNOW

[Figure]

**Step 4:** Distribute %SNOW over the 4 km PRISM grid points (binned by 200 m intervals) using bilinear interpolation. For seasonal values, use PRISM P to scale %SNOW

[Figure]

*Figure 2: Conceptual diagram illustrating the four key steps in the calculation of %SNOW at 4 km horizontal resolution and using 200 m elevation bins starting with 2.5° x 2.5° horizontal resolution NCEP/NCAR reanalysis.*

Minor Concerns
One thing that is missing from the introduction and could be a nice addition is an understanding of how this indicator could actually be used for water resource planning. This is touched upon in the Discussion (Page 5), but I think some of this type of information needs to be included in the introduction to further motivate the need for this method. You do say that changes in rain-snow partition are important for water storage –however it wasn't until the Discussion that I could really see why this indicator might actually be used for planning purposes.

This is a valuable suggestion, especially for a methods/technical note type paper intended to help other practitioners/managers as well as the science community. We have revised and added text to the introduction to better describe some of the specific ways that DWR is applying this information. For the most part, they use it as an indicator of change for situational awareness of where expected impacts of climate change are occurring (and how fast). We also added a final sentence noting the approach can be used to look forward as well (longer-term planning).

New text:
"***Since 2015, DWR has documented this indicator in its annual Hydroclimate Report (DWR, 2019). Though not used directly in operational forecasts, the indicator provides DWR with situational awareness regarding the location of changes in precipitation phase and the rates of these changes.*** Because the method uses publicly available gridded data sets, the indicator is scalable to regional-to-continental scales and therefore could be an informative diagnostic for water resources management and model development in snowmelt dependent regions. While we focus on California watersheds, an example application to the western United States is provided. ***Last, instead of historic data, it can also use model projections as input to help inform the development of adaptation strategies to achieve water resource management goals amidst a changing climate.***"

Page 1, Line 23: "components has been used as a foundation"

Thank you for suggesting a fix, in light of other reviewer comments we revised the sentence as follows:

"*The partitioning of precipitation into liquid (rain) and, in particular, frozen (snow) components along with climatic stationarity were foundational assumptions in the development of water management infrastructure and practices in California and other mountain environments in the western United States (US) since the mid-1800s (Milly et al., 2008).*"

Page 1, Line 27: The use of the word "fate" is a little awkward here, do you mean "phase"?

We agree 'fate' is awkward, though the original intent was a meaning of fate as "destiny", where the destiny of cool season precip is either rain that runs off or snow that accumulates. We

changed to use the suggestion of "phase" as this is less awkward and is correct, as the phase does ultimately drive the management strategy.

New text:
"The **phase** of cool season precipitation ultimately drives"

Page 1, Line 36: Unclear what you mean by "winter snow levels" here. Do you mean the freezing level in the atmosphere or do you mean increase snow pack (which would be counter intuitive). The jargon of 'winter snow levels" is confusing.

We understand the reviewer's confusion and appreciate them pointing this out as they are not the first to be confused by the nomenclature. We changed the text to winter snow line elevation (which is close to the freezing level, but usually several hundred meters below due to melting times/other processes that influence melting rates).

New text: "winter snow **line elevation**"

Page 2, Lines 9-11: The first sentence of this paragraph is not complete. It is unclear what are you incorporating multiple data sources and model outputs into.

We appreciate the suggestion to correct the structure of this sentence. We have revised it accordingly to be clearer that the problem is little data availability to apply hydrologic models or to evaluate change. Incorporation of multiple data sources or other model output can often provide the necessary pieces to perform the study of interest.

New text:
"The sparse observational networks and complex topography of the western US introduces challenges into basin-scale hydrologic monitoring and modelling***. To address these challenges when applying hydrologic models or for monitoring long-term change,*** the incorporation of multiple sources of data (Bales et al., 2006) and/or model output (e.g., Wrzesian et al., 2019) ***is often required.***"

You mention on Page 2, Line 14 that DWR developed a methodology to study historical rain/snow trends at fine spatial resolutions and then on Page 2, Line 16 that the purpose of the note is to provide an updated approach and detail the methods of this indicator. It is unclear to me what in this paper is from the original DWR method and what is the "updated" approach. Is the only difference between the DWR method and the method described in the paper the resolution of the PRISM model data? If the goal of the paper is just to outline the DWR approach –then state that and remove the confusing "updated approach" language. However if the DWR approach is documented elsewhere, and you are documenting changes here in this paper, those differences need to be more explicitly stated.

We apologize for the confusing language. This document is intended to provide the first peer-reviewed documentation of the approach originally described in the 2014 DWR report. We have removed 'updated' to avoid confusion.

New text:
"The purpose of this technical note is to *describe the development of this diagnostic indicator aimed at quantifying how rain and snow are partitioned.*"

The reviewer is correct that the only difference described here is with regards to the PRISM spatial resolution. Because this is already described in the methods (section 2.2; the DWR approach uses 800 m but because of data accessibility (free vs. pay) we use the 4 km product), we chose not to further explain this difference in the introduction.

Page 2, Line 16: You say "detail the methods of this indicator" but at this point in the paper it is not clear what "indicator" you are talking about. A sentence about the "indicator" before this one is needed. (e.g. Page 6, Line 4 could also be stated here in the text).

Thank you for the request for additional clarity. We revised the text as follows:
"…indicator **of how rain and snow are partitioned**."

Page 2, Line 19: "and be an important" a modal verb is need before "be" as in "may be" or "can be" etc.
We appreciate the grammatical correction (and example!). The text now reads:
"and **therefore could** be an *informative*"

Page 3, Line 20: ...the percent of precipitation that falls (as snow??) at elevations above the 0ºisotherm ...
Thank you for pointing out our omission. Text has been changed:
"…falls **as snow** at…"

2.1 the Study Area Did DWR create this method exclusively for studying trends in rain/snow partitioning, or is this data used in operational forecasts?

Correct, DWR did develop the method exclusively for studying trends. We revised to make it clear in the introduction that the indicator is not used operationally but to inform about trends:
"***Though not used directly in operational forecasts***, *the indicator provides DWR with situational awareness regarding the location of changes in precipitation phase and the rates of these changes.*"

Figure 4 has a number of problems:
What season is being plotted? Entire water year? Cold season etc?

Thank you for pointing out this omission. These are water year plots. The text has been changed to specify this (note this becomes Figure 5 since we added the suggested conceptual diagram):

"Figure 5: (a) Aggregated trends in %$_{SNOW}$ (% decade$^{-1}$) by latitude and elevation ***for the water year.*** Dot size is scaled by area of watershed occupying each elevation and latitude bin. (b) Elevation-based trends (aggregated over all latitudes) of %$_{SNOW}$ (% decade$^{-1}$) showing median (black line), the interquartile range (dark grey shading), and 90% confidence intervals (light grey shading) on the left y-axis. Right y-axis shows the total watershed area occupied by each elevation bin (red line; km$^2$). Aggregations were performed on gridpoints within the subset of California Department of Water Resources analysis zones (see Figure 1a)."

You discuss Figure 5b before 5a, they should be flipped in the panel.
Thank you for pointing this out, we flipped the panels (and adjusted the caption order as well). New figure and caption:

[Figure]

Figure 5: *(a) Elevation-based trends (aggregated over all latitudes) of %$_{SNOW}$ (% decade$^{-1}$) showing median (black line), the interquartile range (dark grey shading), and 90% confidence intervals (light grey shading) on the left y-axis. Right y-axis shows the total watershed area occupied by each elevation bin (red line; km$^2$). (b) Aggregated trends in %$_{SNOW}$ (% decade$^{-1}$) by*

*latitude and elevation for the water year. Dot size is scaled by area of watershed occupying each elevation and latitude bin.* Aggregations were performed on gridpoints within the subset of California Department of Water Resources analysis zones (see Figure 1a) and sorted by elevation. The interquartile range (IQR) and 90% confidence interval (CI) were estimated using all grid points within each elevation band and analysis zone.

It is unclear from the text how Figure 4b is calculated –what is the IQR and 90% CI based on (is this covering every grid point within that elevation band?).

Correct, and we apologize for the oversight to include this detail in our original submission. We added a sentence to the caption to describe how the IQR and CI were calculated (please also see complete revised caption above):
 *"The interquartile range (IQR) and 90% confidence interval (CI) were estimated using all grid points within each elevation band and analysis zone."*

Does Figure A represent the values shown in Figure 3 but sorted by elevation?

Yes, the difference being values were aggregated by latitude and elevation and not by watershed. We added a note in the Figure 5 (previously Fig 3) caption that the aggregations were *"sorted by elevation"*.

Page 5, Line 37 –what is a flood pool? This should be stated in a way that non-flood forecasters/water managers can understand.

Good suggestion. We added the definition and some text clarifying why it matters. The flood pool exists to prevent downstream flooding when inflows are high (such as during storms). The states that a flood pool must be maintained during certain parts of the year, meaning that inflows into the flood pool must be released as soon as possible. Instead of water being stored upstream in the snowpack to flow into the reservoir in July (a resource), now this water can be lost for later consumptive use as it flows downstream in February (managed as a hazard).

New text:
"More precipitation falling as rain during storms, especially in regions with large watershed areas in lower elevations, increases *midwinter* inflow into reservoirs. Many current multipurpose reservoir management paradigms require the maintenance of a flood pool, *which is reservoir storage space allocated to attenuate periods of heavy inflow and reduce flood hazard during cool season storms. Water captured during the flood is later released to maintain the flood pool storage capabilities during the next possible event. Flood pool releases* mean this water cannot be stored for later beneficial use and must be managed as a hazard rather than a resource."

Page 6, Line 4 –this description of the goal of this paper needs to be moved up into the introduction.

Thank you for the suggestion. We moved the sentence to the introduction.

[revised manuscript text omitted]

---

## Author Response (AR2)

**Dear Dr. Viviroli,**
**Please find our technical corrections following re-review by Reviewer #2 below for Lynn et al. (2020): "Technical Note: Precipitation phase partitioning at landscape-to-regional scales".**
**Thank you for your assistance with this manuscript.**
**Cheers,**
**Benjamin Hatchett**

Author comments provided in blue. Changes are given in *italics with **bold for emphasis.***

Re-review of "Technical Note: Precipitation phase partitioning at landscape-to-regional scales" By Lynn et al. (2020)

The authors have done an outstanding job responding to my reviewer comments and the comments of the other reviewer. They have substantially improved this technical note and I feel it is ready for publication following some very minor technical fixes.

My primary concerns were regarding ambiguity in their description of the method and their use of NCEP/NCAR Reanalysis 1. The work they have done to explore precipitation phasing in ERA5, their addition of a visual schematic of their methods (Figure 2) and overall improved clarity of the writing of the methodology section has more than satisfied my concerns.

We greatly appreciate your kind words! We again thank the reviewer for their constructive comments from which the paper benefitted greatly.

Here are some very minor technical problems I found while reading the paper:

Page 3 Line 18: there needs to be a comma between isotherm and across.

Comma has been added.

Page 3 lines 22-24: these are very important characteristics of the role the freezing level play s- are they from the Diaz paper? This is unclear and I think this needs a reference.

Thank you for the suggestion. Yes, these are from the Diaz paper but we also added the White et al., 2010 paper and the following two references to the sentence to provide a more complete list:

Contosta, A. R., Casson, N. J., Garlick, S., Nelson, S. J., Ayres, M. P., Burakowski, E. A., Campbell, J., Creed, I., Eimers, C., Evans, C., Fernandez, I., Fuss, C., Huntington, T., Patel, K., Sanders-DeMott, R., Son, K., Templer, P., and Thornbrugh, C.: Northern forest winters have lost cold, snowy conditions that are important for ecosystems and human communities, *Ecol. Appl.,* 29(7):e01974, doi:10.1002/eap.1974, 2019.

Sospedra-Alfonso, R., Melton, J. R., and Merryfield, W. J.: Effects of temperature and precipitation on snowpack variability in the Central Rocky Mountains as a function of elevation. *Geophys. Res. Lett.*, 42, 4429–4438. doi: 10.1002/2015GL063898, 2015.

Page 3 line 29: I think this sentence would read better if flipped: "For cases in which the vertical temperature profile includes inversion conditions with multiple incursions of the 0º isotherm, the uppermost atmospheric level below which the 0ºC isotherm occurs is used.

We agree this reads more intuitively in the manner suggested by the reviewer. We have revised the sentence following the reviewer's suggestion:
*"For cases in which the vertical temperature profile includes inversion conditions with multiple incursions of the 0°C isotherm, the uppermost atmospheric level below which the 0°C isotherm occurs is used."*

Page 7 Line 5: Clarify "observational data" with station or insitu observational data, as satellite observations and other methods of measuring precipitation are not included in the development of PRISM.

Good point, this change has been made:
"…method based on ***in situ*** observational data…"